# SpaFL: Communication-Efficient Federated Learning with Sparse Models and Low Computational Overhead

## Abstract

The large communication and computation overhead of federated learning (FL) is one of the main challenges facing its practical deployment over resource-constrained clients and systems. In this work, SpaFL: a communication-efficient FL framework is proposed to optimize both personalized model parameters and sparse model structures with low computational overhead. In SpaFL, a trainable threshold is defined for each neuron/filter to prune its connected parameters. Both model parameters and thresholds are jointly optimized to enable the automatic sparsification of the models while recovering prematurely pruned parameters during training. To reduce communication costs, only thresholds are communicated between a server and clients instead of parameters, thereby enabling the clients to learn how to prune. Further, global thresholds are used to update model parameters by extracting aggregated parameter importance. The convergence of SpaFL is analyzed, and the results provide new insights into the tradeoff between computation costs and learning performance. Experimental results show that SpaFL improves accuracy while requiring much less communication and computing resources compared to both dense and sparse personalized baselines.

## 1 Introduction

Federated learning (FL) is a distributed machine learning framework in which clients collaborate to train a machine learning (ML) model without sharing private data [1]. In FL, clients perform multiple epochs of local training using their own datasets and communicate model updates with a server. Different from a standard centralized setting, FL systems are typically deployed on edge devices such as mobile or Internet of Things (IoT) devices, which have limited computing and communication resources. However, current ML models are typically too large and complex to be trained and deployed for inference by edge devices. Moreover, large model sizes can induce significant FL communication costs on both devices and communication networks. Hence, the practical deployment of FL over *resource-constrained devices and systems* requires optimized computation and communication costs for both edge devices and communication networks. This has motivated lines of research focused on reducing communication overhead in FL [2, 3, 4], training sparse neural networks in FL [5, 6, 7, 8, 9], and optimizing model architectures to find a compact model for inference [10, 11, 12]. The works in [2, 3, 4] proposed training algorithms such as quantization, gradient compression, and transmitting the subset of models in order to reduce the communication costs of FL. However, the associated computational overhead of these existing algorithms remains high since devices have to train a dense model. In [5, 6, 7, 8, 9], FL algorithms in which devices train and communicate sparse models are proposed. However, the computation and communication overhead can still be large if model sparsity is not high. Moreover, the model performance often becomes low for high model sparsity. Furthermore, the FL approaches of [10, 11, 12] can significantly increase computation resource usage by training multiple models for resource-constrained devices. Clearly, despite a surge of literature on sparsity in FL, there is still a need to develop new FL algorithms that can obtain sparse models with optimized communication efficiency and low computational overhead to operate on resource-constrained devices while maintaining model performance with high sparsity.

The main contribution of this paper is to propose *SpaFL: a communication-efficient FL framework for optimizing sparse models with low computational overhead* by performing model pruning through

trainable thresholds. SpaFL communicates only the thresholds so as to learn how to prune and significantly save communication costs. Here, a trainable threshold is defined for each neuron/filter and is used to prune its connected parameters based on magnitude. Both parameters and thresholds are jointly optimized to enable the automatic sparsification of models while recovering prematurely pruned parameters during training. Therefore, SpaFL reduces computational overhead on resource constrained devices with minimal performance loss. To further save communication costs, *only thresholds are communicated* between clients and a server. Hence, clients can learn how to prune their model from global thresholds. Since parameters are not communicated, the clients' parameters and sparse model structures will remain personalized while only global thresholds are shared. We show that global thresholds can capture the aggregated parameter importance of clients. We further update the clients' model parameters by extracting aggregated parameter importance from global thresholds to improve performance. We analyze the convergence of thresholds while shedding light on the tradeoff between computational overhead and performance. We summarize our contributions as follows:

- We propose a new communication-efficient FL framework called SpaFL, in which clients optimize their personalized model parameters and sparse model structures with low computing costs through trainable thresholds.

- We show how SpaFL can significantly reduce communication overhead for both clients and the server by only exchanging thresholds, the number of which is less than two orders of magnitude smaller than the number of model parameters.

- We prove the convergence of thresholds. Moreover, the impact of thresholds on the model performance is theoretically and experimentally analyzed.

- Experimental results demonstrate the performance, computation costs, and communication efficiency of SpaFL compared with both dense and sparse baselines. For instance, the results show that SpaFL uses only 2.87% of communication and 14.7% of computation resources compared to a dense baseline FedAvg while improving accuracy. Additionally, SpaFL improves accuracy by 2.13% compared to a sparse personalized baseline while consuming only 25.55% of this baseline's communication resources, and only 14.67% of its computing resources.

## 2 BACKGROUND AND STATE-OF-ART

### 2.1 PERSONALIZED FL

Personalized FL is a field of FL whose goal is to produce personalized models for each client to cope with the data heterogeneity. Personalization can be done via multiple methods such as layer personalization [2, 13, 14], regularization term [15, 16], fine-tuning [17, 18], and knowledge distillation [19, 20]. Although personalization can effectively tackle non-iid datasets, it requires extra computation [15, 17, 18, 19, 20] for resource-constrained clients, and its communication overhead can also be high due to large model sizes. As such, computation and communication costs are still one of the major challenges for personalized FL. This has motivated a line of research in training sparse personalized models [7, 9, 21, 22, 23, 24] by allowing clients to locally prune a global model during training.

### 2.2 TRAINING AND FINDING SPARSE MODELS IN FL

To reduce the computation and communication overhead of complex ML models during training, the idea of embedding FL algorithms with pruning has recently emerged. In [5, 6, 7, 8, 9, 21, 22, 23, 24, 25, 26, 27], the clients train sparse models and only communicate sparse model parameters to reduce computation and communication overhead. To improve the aggregation phase with sparse models, the works in [5, 8, 21] perform averaging only between overlapping parameters to avoid information dilution by excluding zero value parameters. The work in [6] obtained a sparse model by selecting a particular client to prune an initial dense model and then performed training in a similar way to FedAvg. In [25], the authors presented binary masks adjustment strategy to improve the performance of sparse models and communication efficiency. The work in [26] progressively pruned a dense model for sparsification and analyzed its convergence. Similarly, in the FL solutions of [9, 21], the clients

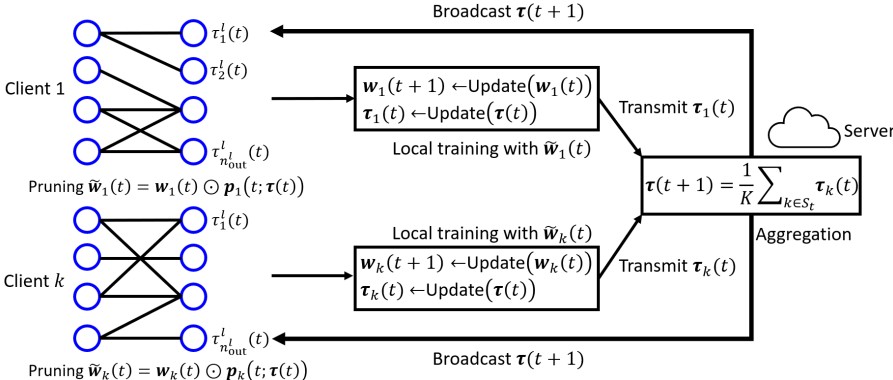

Figure 1: Illustration of SpaFL framework that performs model pruning through thresholds. Only the thresholds are communicated between the server and clients.

train and communicate personalized sparse local models while iteratively pruning a dense model. In [7, 22], the clients optimized personalized sparse models by exchanging lottery tickets [28] at every communication round. The work in [24] obtained personalized sparse models by $l_1$ norms constraints and the correlation between local and global models. The FL framework of [23] allows clients to train personalized sparse models in a decentralized setting without a central server. Although these works adopted sparse models during training, if models are not sparse enough, the computation and communication costs can remain high due to the large model sizes. Moreover, trained models often have a poor performance at high sparsity. In [29, 30], binary masks are communicated and optimized by training auxiliary variables while freezing model parameters. However, the work in [29] approximated binarization step using a sigmoid function during forward propagation. In [30], the downlink communication costs remained the same as that of FedAvg. In [10, 11, 31], clients perform neural-architecture-search by training multiple models to find optimized and sparse models to improve computational and memory efficiency at inference phase. However, in practice, clients often have limited resources to support the computationally intensive architecture search process. Therefore, most prior works either required extensive computational resources to find sparse models, or achieved a poor performance for high model sparsity. Different from prior works, in the proposed SpaFL framework, we do not incur more computational resources to find sparse models nor do we jeopardize the performance for sparsification.

## 3 SpaFL Algorithm

In this section, we first formulate our FL problem and present SpaFL to solve the proposed problem with low computation and communication costs.

### 3.1 Problem Formulation

We now formulate our main problem. We aim to optimize each client's model parameters and sparse model structures jointly in a personalized FL setting. This can be formulated as the following bi-level optimization problem:

$$\min_{\boldsymbol{p}_1,\ldots,\boldsymbol{p}_N} \quad \sum_{k=1}^{N} F_k(\boldsymbol{w}_k^* \odot \boldsymbol{p}_k; \boldsymbol{w}_k^*),$$

$$\text{s.t.} \quad \boldsymbol{w}_k^* = \arg\min_{\boldsymbol{w}} F_k(\boldsymbol{w} \odot \boldsymbol{p}_k; \boldsymbol{p}_k),$$

$$F_k(\boldsymbol{w} \odot \boldsymbol{p}_k; \boldsymbol{p}_k) = \frac{1}{D_k} \sum_{i=1}^{D_k} \mathcal{L}(\boldsymbol{w} \odot \boldsymbol{p}_k; \{\boldsymbol{x_i}, y_i\}) \qquad (1)$$

where $\boldsymbol{p}_k \in \mathbb{R}^d$ is a binary mask corresponding to a personalized sparse model structure of client $k$, $F_k(\cdot)$ is a empirical risk associated with local data of client $k$, $\mathcal{L}$ is a loss function, $D_k$ is the number of data samples, $\{\boldsymbol{x}, y\}$ is an input-label pair, $\boldsymbol{w}_k$ captures the model parameters, and $\odot$ is the Hadamard product. If the element of $\boldsymbol{p}_k$ is zero, then the corresponding parameter of $\boldsymbol{w}_k$

will be pruned. Our goal is to obtain the optimal $\boldsymbol{w}_k$ and $\boldsymbol{p}_k$ for each client in order to reduce the computation and communication overhead during training while being robust to the data heterogeneity among clients. However, solving (1) is not trivial because $\boldsymbol{w}_k$ and $\boldsymbol{p}_k$ are highly correlated [11]. Moreover, personalization and model sparsification should be achieved in a resource-efficient manner because most clients (e.g., mobile or IoT devices) and communication systems do not have enough computation and communication resources.

## 3.2 Algorithm Overview

We now describe the proposed algorithm, SpaFL, that can solve (1) while maintaining communication-efficiency with low computational cost. In SpaFL, every client jointly optimizes its personalized sparse model structure and model parameters with trainable thresholds, which can be used to prune the model parameters. To save communication resources, only thresholds will be aggregated at a server to generate global thresholds for the next round. Here, global thresholds can represent the aggregated parameter importance of clients. Hence, at the beginning of each round, every client extracts the aggregated parameter importance from the global thresholds so as to update its model parameters. The overall algorithm is illustrated in Fig 1. and summarized in Algorithm 1.

### 3.2.1 Pruning with trainable thresholds

To optimize the binary mask $\boldsymbol{p}_k$ in (1), inspired from [32], we define a trainable threshold for each neuron in fully-connected layers or for each filter in convolutional layers. The neural network of client $k$ will consist of $L$ layers as $\{\boldsymbol{W}_k^1, \ldots, \boldsymbol{W}_k^L\}$. For parameters $\boldsymbol{W}_k^l \in \mathbb{R}^{n_{\text{out}}^l \times n_{\text{in}}^l}$ in a fully-connected layer $l$, we define trainable thresholds $\boldsymbol{\tau}^l \in \mathbb{R}^{n_{\text{out}}^l}$. If it is a convolutional layer $\boldsymbol{W}_k^l \in \mathbb{R}^{n_{\text{out}}^l \times c_{\text{in}}^l \times k^l \times h^l}$, where $c_{\text{in}}^l$ is the number of input channels and $k^l \times h^l$ are the kernel sizes, we can change $\boldsymbol{W}_k^l$ as $\boldsymbol{W}_k^l \in \mathbb{R}^{n_{\text{out}}^l \times n_{\text{in}}^l}$ with $n_{\text{in}}^l = c_{\text{in}}^l \times k^l \times h^l$. Similarly, we can define the corresponding thresholds $\boldsymbol{\tau}^l \in \mathbb{R}^{n_{\text{out}}^l}$. Then, for each client $k$, we define a set of total thresholds $\boldsymbol{\tau} = \{\boldsymbol{\tau}^1, \ldots, \boldsymbol{\tau}^L\}$. Note that the number of these additional thresholds will be at most 1% of the number of model parameters $d$. Moreover, to further improve the communication efficiency, only these thresholds will be communicated between clients and the server.

We prune a parameter if its magnitude is smaller than its connected neuron/filter's threshold. Then, we can obtain a binary mask $\boldsymbol{p}_k^l$ for $\boldsymbol{W}_k^l$, as follows

$$p_{k,ij}^l = S(|w_{k,ij}^l| - \tau_i^l),\ 1 \le i \le n_{\text{out}}^l, 1 \le j \le n_{\text{in}}^l, \tag{2}$$

where $S(\cdot)$ is a unit step function. Hence, we can obtain the binary masks $\{\boldsymbol{p}_k^1, \ldots, \boldsymbol{p}_k^L\}$ by performing (2) at each layer. To facilitate the pruning, we constrain the parameters and thresholds to be within $[-1, 1]$ and $[0, 1]$, respectively [32]. For simplicity, we unroll $\{\boldsymbol{W}_k^1, \ldots, \boldsymbol{W}_k^L\}$ and $\{\boldsymbol{p}_k^1, \ldots, \boldsymbol{p}_k^L\}$ to $\boldsymbol{w}_k \in \mathbb{R}^d$ and $\boldsymbol{p}_k \in \mathbb{R}^d$, respectively as done in [33].

### 3.2.2 Local Training for Parameters and Thresholds

At each round, a server samples a set of clients $\mathcal{S}_t$ such that $|\mathcal{S}_t| = K$ for local training. For given global thresholds $\boldsymbol{\tau}(t)$ at round $t$, client $k \in \mathcal{S}_t$ generates a binary mask $\boldsymbol{p}_k(t; \boldsymbol{\tau}(t))$ using (2). Subsequently, it obtains the sparse model $\tilde{\boldsymbol{w}}_k(t) = \boldsymbol{w}_k(t) \odot \boldsymbol{p}_k(t; \boldsymbol{\tau}(t))$. To improve communication efficiency, each client performs $E$ epochs with $\boldsymbol{p}_k(t; \boldsymbol{\tau}(t))$. In particular, for $E > 1$, client $k \in \mathcal{S}_t$ performs $E - 1$ epochs to its sparse model using mini-batch stochastic gradient as follows:

$$\boldsymbol{w}_k^{e+1}(t) \leftarrow \boldsymbol{w}_k^e(t) - \eta(t) \boldsymbol{g}_k(\tilde{\boldsymbol{w}}_k^e(t)) \odot \boldsymbol{p}_k(t; \boldsymbol{\tau}(t)),\ 0 \le e < E - 1, \tilde{\boldsymbol{w}}_k^0(t) = \tilde{\boldsymbol{w}}_k(t), \tag{3}$$

where $\boldsymbol{g}_k(\tilde{\boldsymbol{w}}_k^e(t)) = \nabla_{\tilde{\boldsymbol{w}}_k^e} F_k(\tilde{\boldsymbol{w}}_k^e(t), \boldsymbol{\tau}(t); \xi_k^e(t))$ with a mini-batch $\xi$ and $\eta(t)$ is a learning rate for the model parameters. During $E - 1$ epochs, the model parameters can adapt to the new binary mask $\boldsymbol{p}_k(t; \boldsymbol{\tau}(t))$, which is generated using the current global thresholds $\boldsymbol{\tau}(t)$, by recovering from pruning-induced noise [34]. Then, in the last epoch $e = E - 1$, client $k$ updates the received global thresholds $\boldsymbol{\tau}(t)$ via backpropagation. Client $k$ first calculates the following sparsity regularization term $R(t) = \sum_{l=1}^L \sum_{i=1}^{n_{\text{out}}^l} \exp(-\tau_i)$. Then, the loss function at the last epoch $e = E - 1$ will be:

$$F_{\boldsymbol{\tau},k}(t) = F_k(\tilde{\boldsymbol{w}}_k^{E-1}(t), \boldsymbol{\tau}(t); \xi_k^{E-1}(t)) + \alpha R(t), \tag{4}$$

where $0 \leq \alpha \leq 1$ is the coefficient that controls $R(t)$. From (4), we can give thresholds $\boldsymbol{\tau}(t)$ performance feedback on the current sparse model while also progressively increasing $\boldsymbol{\tau}(t)$ through the sparsity regularization term $R(t)$ [32]. Hence, prematurely pruned parameters can be recovered since thresholds and parameters are jointly updated while progressively enforcing sparsity. From (4), client $k$ then updates the received global thresholds $\boldsymbol{\tau}(t)$ via backpropagation as follows

$$\tau_{k,i}^l(t) \leftarrow \tau_i^l(t) - \eta(t)h_{k,i}^l(\tilde{\boldsymbol{w}}_k^{E-1}(t)) + \alpha\eta(t)\tau_i^l(t), 1 \leq l \leq L, 1 \leq i \leq n_{\text{out}}^l, \tag{5}$$

where

$$h_{k,i}^l(\tilde{\boldsymbol{w}}_k^{E-1}(t)) = -\sum_{j=1}^{n_{\text{in}}^l}\{\boldsymbol{g}_k(\tilde{\boldsymbol{w}}_k^{E-1}(t))\}_{ij}^l w_{k,ij}^{E-1,l}(t). \tag{6}$$

For simplicity, we can vectorize (5) as follows:

$$\boldsymbol{\tau}_k(t) = \boldsymbol{\tau}(t) - \eta(t)\boldsymbol{h}_k(\tilde{\boldsymbol{w}}_k^{E-1}(t)) + \alpha\exp(-\boldsymbol{\tau}(t)), \; ||\boldsymbol{h}_k(\tilde{\boldsymbol{w}}_k^{E-1}(t))||^2 = \sum_{l=1}^{L}\sum_{i=1}^{n_{\text{out}}^l}||h_{k,i}^l(\tilde{\boldsymbol{w}}_k^{E-1}(t))||^2, \tag{7}$$

where $\boldsymbol{h}_k(\tilde{\boldsymbol{w}}_k^{E-1}(t)) = \nabla_{\boldsymbol{\tau}}F_k(\tilde{\boldsymbol{w}}_k^{E-1}(t), \boldsymbol{\tau}(t); \xi_k^{E-1}(t))$. When we calculate the gradients of thresholds $\boldsymbol{\tau}(t)$, we use the identity straight-through estimator [35] to approximate the gradient of the step functions in (2).

From (6), we can see that threshold $\tau_{k,i}^l$ corresponds to the importance of its connected parameters $w_{k,ij}^l, 1 \leq j \leq n_{\text{in}}^l$, in the sparse model $\tilde{\boldsymbol{w}}_k(t)$. This is because the importance of a parameter $w_{ij}^l$ can be estimated by [36]

$$F(\boldsymbol{w}, \boldsymbol{\tau}) - F(\boldsymbol{w}, \boldsymbol{\tau}; w_{ij}^l = 0) \approx g(\boldsymbol{w})_{ij}^l w_{ij}^l, \tag{8}$$

where $F(\boldsymbol{w}, \boldsymbol{\tau}; w_{ij}^l = 0)$ is the loss function when $w_{ij}^l$ is masked and the approximation is obtained from the first Taylor expansion at $w_{ij}^l = 0$. Therefore, if connected parameters were important, the sign of (8) of those parameters will be negative, and the corresponding threshold will decrease as in (6). Otherwise, the threshold will be increased to enforce sparsity. Hence, prematurely pruned parameters will be automatically recovered via a joint optimization of $\boldsymbol{\tau}$ and $\boldsymbol{w}$.

### 3.2.3 COMMUNICATION-EFFICIENT THRESHOLDS TRANSMISSION AND AGGREGATION

After local training, each client $k \in \mathcal{S}_t$, transmits the updated thresholds $\boldsymbol{\tau}_k(t)$ to the server. Here, the communication overhead will be less than one percent of that of transmitting the entire parameters. Subsequently, the server aggregates the received thresholds and generates new global thresholds for the next round, i.e.,

$$\boldsymbol{\tau}(t+1) = \frac{1}{K}\sum_{k \in \mathcal{S}_t}\boldsymbol{\tau}_k(t). \tag{9}$$

Since thresholds represent the importance of the connected parameters at each neuron/filter, clients can learn how to prune their parameters from the global thresholds. Moreover, the difference between two consecutive global thresholds $\Delta\boldsymbol{\tau}(t) = \boldsymbol{\tau}(t+1) - \boldsymbol{\tau}(t)$ captures the history of aggregated parameter importance, which can be further used to improve model performance. For instance, from (8), if $\Delta\tau_i^l(t) < 0$, then the parameters connected to threshold $i$ in layer $l$ were globally important. If $\Delta\tau_i^l(t) \geq 0$, then the connected parameters were globally less important. Hence, from $\Delta\boldsymbol{\tau}(t)$, clients can deduce which parameter is globally important or not and further update their model parameters. After generating new global thresholds $\boldsymbol{\tau}(t+1)$, the server broadcasts $\boldsymbol{\tau}(t+1)$ to client $k \in \mathcal{N}$, and then clients calculate $\Delta\boldsymbol{\tau}(t) = \boldsymbol{\tau}(t+1) - \boldsymbol{\tau}(t)$.

### 3.2.4 EXTRACTING PARAMETER IMPORTANCE FROM GLOBAL THRESHOLDS

We now present how clients can update their model parameters from $\Delta\boldsymbol{\tau}(t)$. For given $\Delta\boldsymbol{\tau}(t)$, we need to decide on the: 1) update direction and 2) update amount. Clients can know the update direction of parameters by considering $\Delta\boldsymbol{\tau}(t)$ and the dominant sign of parameters connected to

---

**Algorithm 1:** SpaFL

---

**Input:** Total number of clients $N$; Total communication rounds $T$; Local number of epochs $E$
**Output:** Personalized models $\tilde{\boldsymbol{w}}_k$

1   The server initializes $\boldsymbol{\tau}(0)$ and $\boldsymbol{w}(0)$ and broadcasts them to every client ;
2   **for** $t = 0$ *to* $T - 1$ **do**
3      Server randomly samples $\mathcal{S}_t$;
4      **for** *Client* $k \in \mathcal{S}_t$ **do**
5          Generate a binary mask $\boldsymbol{p}_k(t; \boldsymbol{\tau}(t))$ and prune the current model $\tilde{\boldsymbol{w}}_k(t) = \boldsymbol{w}_k(t) \odot \boldsymbol{p}_k(t; \boldsymbol{\tau}(t))$;
6          **for** $e = 0$ *to* $E - 1$ **do**
7              **if** $e < E - 1$ **then**
8              Update $\boldsymbol{w}_k^e(t) \leftarrow \tilde{\boldsymbol{w}}_k^e(t) - \eta(t)\boldsymbol{g}_k(\tilde{\boldsymbol{w}}_k^e(t)) \odot \boldsymbol{p}_k(t; \boldsymbol{\tau}(t))$
9              **else if** $e == E - 1$ **then**
10              Update $\boldsymbol{\tau}(t)$ using (5)
11          Transmit the updated threshold $\boldsymbol{\tau}_k(t)$ to the server
12      Generate a new global threshold $\boldsymbol{\tau}(t+1)$ using (9)
13      **for** *Client* $k \in N$ **do**
14          Receive $\boldsymbol{\tau}(t+1)$ from the server and calculate $\Delta\boldsymbol{\tau}(t)$;
15          Update the current local model using $\Delta\boldsymbol{\tau}(t)$ with (12);

---

each threshold. For simplicity, assume that each parameter has a threshold. Then, the gradient of the thresholds in (6) can be rewritten as follows:

$$\boldsymbol{h}_k(\tilde{\boldsymbol{w}}_k^{E-1}(t)) = -\boldsymbol{g}_k(\tilde{\boldsymbol{w}}_k^{E-1}(t))\boldsymbol{w}_k(t). \tag{10}$$

The gradient of the loss $F_k(\tilde{\boldsymbol{w}}_k(t), \boldsymbol{\tau}(t))$ with respect to the whole parameters $\boldsymbol{w}_k(t)$ is given by

$$\frac{\partial F_k(\tilde{\boldsymbol{w}}_k(t), \boldsymbol{\tau}(t))}{\partial \boldsymbol{w}_k(t)} = \boldsymbol{g}_k(\tilde{\boldsymbol{w}}_k(t))|\boldsymbol{w}_k(t)|. \tag{11}$$

From (10) and (11), the gradient direction of a parameter $w$ is opposite of that of its connected threshold if $w > 0$. Otherwise, both the threshold and the parameter have the same gradient direction. Hence, we can deduce the following: If $w > 0$, the gradient direction of $w$ and the sign of $\Delta\tau$ will have the same sign; otherwise, the gradient direction of $w$ and the sign of $\Delta\tau$ are opposite. Since in SpaFL each threshold has multiple connected parameters, we decide the update direction of connected parameters by finding the dominant sign among them. To this end, we simply add the connected parameters of each threshold. For instance, consider threshold $i$ in layer $l$ of client $k$, if $\sum_{j=1}^{n_{\text{in}}^l} w_{k,ij}^l(t) > 0$, then the gradient direction of the connected parameters will be the same as the sign of $\Delta\tau_i^l(t)$. Otherwise, it is the opposite of the sign of $\Delta\tau_i^l(t)$. Thus, the update direction can be simply expressed with a XOR operation between the sign of $\Delta\tau_i^l(t)$ and the sign of connected parameters sum. Next, we decide how much a parameter should be updated. From (10) and (11), we can see that a threshold and a parameter have the same magnitude for their gradients. Hence, we simply divide $\Delta\tau_i^l(t)$ by the number of connected parameters $n_{\text{in}}^l$. We finally provide the update equation using $\Delta\boldsymbol{\tau}(t)$ as follows

$$w_{k,ij}^l(t+1) = w_{k,ij}^l(t) + \frac{1}{n_{\text{in}}^l}\Delta\tau_i^l(t) \text{ XOR}\left\{ \text{sign}\left(\sum_{j=1}^{n_{\text{in}}^l} w_{k,ij}^l(t)\right), \text{sign}(\Delta\tau_i^l(t))\right\}, \tag{12}$$

where $\text{sign}(\cdot)$ is a sign function. This parameter update corresponds to line 7 in Algorithm 1. Note that this additional parameter update is not computationally intensive because it happens only once before local training. We also provide the number of used FLOPs during training with inclusion of this operation in Section 5.

## 4   THEORETICAL ANALYSIS OF SPAFL

We now present our analysis on the convergence of thresholds under the following commonly adopted assumptions [37].

**Assumption 1.** *(smoothness)* $F_k(\cdot)$ *is $M$-smooth for $\boldsymbol{\tau}$ and client $k$, $\forall k$*

$$F_k(\boldsymbol{w}, \boldsymbol{\tau}') \leq F_k(\boldsymbol{w}, \boldsymbol{\tau}) + \langle \nabla_{\boldsymbol{\tau}} F_k(\boldsymbol{w}, \boldsymbol{\tau}), \boldsymbol{\tau}' - \boldsymbol{\tau} \rangle + \frac{M}{2} ||\boldsymbol{\tau}' - \boldsymbol{\tau}||^2, \; \forall \boldsymbol{\tau}. \tag{13}$$

**Assumption 2.** *(Unbiased stochastic gradient) The stochastic gradient $\boldsymbol{h}_k$ is an unbiased estimator of the gradient $\nabla_{\boldsymbol{\tau}} F_k$, respectively, for client $k, \forall k$, such that*

$$\mathbb{E} \boldsymbol{h}_k(\boldsymbol{w}_k) = \nabla_{\boldsymbol{\tau}} F_k(\boldsymbol{w}_k, \boldsymbol{\tau}) \tag{14}$$

Then, we can derive the convergence of thresholds in the following theorem.

**Theorem 1.** *For $\gamma(t) = \eta(t)(1 - \frac{\alpha(1 - M\eta(t))}{2})$ and the largest number of parameters connected to a neuron or filter $n_{in}^{max} > 0$ in a given model, we have*

$$\frac{1}{NT} \sum_{t=0}^{T-1} \mathbb{E}|| \sum_{k=1}^{N} \nabla_{\boldsymbol{\tau}} F_k(\tilde{\boldsymbol{w}}_k^{E-1}(t), \boldsymbol{\tau}(t))||^2 \leq \sum_{t=0}^{T-1} \sum_{k=1}^{N} \frac{\mathbb{E}||\nabla_{\boldsymbol{\tau}} F_k(\tilde{\boldsymbol{w}}_k^{E-1}(t), \boldsymbol{\tau}(t)) - \nabla_{\boldsymbol{\tau}_k} F_k(\tilde{\boldsymbol{w}}_k^{E-1}(t), \boldsymbol{\tau}_k(t))||^2}{MNT\gamma(t)}$$

$$+ \sum_{t=0}^{T-1} \frac{2\alpha\eta(t)}{T\gamma(t)} \{1 - M\eta(t)(1 - \alpha)\} || \exp(-\boldsymbol{\tau}(t))||^2$$

$$+ \sum_{t=0}^{T-1} \sum_{k=1}^{N} \frac{M^2\eta(t)^2 n_{in}^{max}}{NT\gamma(t)} \mathbb{E} F_k(\tilde{\boldsymbol{w}}_k^{E-1}(t), \boldsymbol{\tau}(t))$$

$$+ \sum_{t=0}^{T-1} \sum_{k=1}^{N} \frac{\mathbb{E}||\boldsymbol{\tau}(t) - \boldsymbol{\tau}_k(t)||^2}{NT\gamma(t)}. \tag{15}$$

*Proof is provided in the Supplementary document.*

From (6), thresholds $\boldsymbol{\tau}(t)$ are updated using parameter gradients $\boldsymbol{g}_k(t), k \in \mathcal{S}_t$. We can expect that the thresholds will converge when parameters $\boldsymbol{w}_k, \forall k$, converge. We can see that the sparsity regularizer coefficient $\alpha$ impacts convergence. As $\alpha$ increases, we can quickly enforce more sparsity to the model. However, a very large $\alpha$ will damage the performance as $\gamma(t)$ decreases in (**??**). We can also observed that the convergence depends on the difference between the received global thresholds $\boldsymbol{\tau}(t)$ and the updated thresholds $\boldsymbol{\tau}_k(t)$. Hence, a very large change to the global thresholds will lead to a significantly different binary mask in the next round. Then, local training can be unstable as parameters have to adapt to the new mask. Therefore, from Theorem 1, we can capture the tradeoff between the computing cost and the learning performance in terms of $\alpha$.

## 5 EXPERIMENTS

We now present experimental results to demonstrate the performance, computation costs and communication efficiency of SpaFL. Implementation details are provided in the Supplementary document.

### 5.1 EXPERIMENTS CONFIGURATION

We conduct experiments on three image classification datasets: FMNIST [38], CIFAR-10, and CIFAR-100 [39] datasets. To distribute datasets in a non-iid fashion, we use Dirichlet (0.2) for FMNIST and Dirichlet (0.1) for CIFAR-10 and CIFAR-100 datasets as done in [40] with $N = 100$ clients. We set the total communication round $T = 500$. At each round, we randomly sample $K = 10$ clients. Unless stated otherwise, we average all the results over at least 10 different random seeds. We also calculate the best accuracy by averaging each client's performance on its test dataset. For FMNIST dataset, we use the Lenet-5-Caffe and LSTM models. For the Lenet model, we set $\eta(t) = 0.001$, $E = 3$, $\alpha = 0.002$, and a batch size to be 64. For the LSTM model, we use two LSTM layers with hidden size of 128, $\eta(t) = 0.01$, $E = 3$, $\alpha = 0.0003$ and a batch size of 16. For CIFAR-10 dataset, we use a convolutional neural network (CNN) model with seven layers used in [41] with $\eta(t) = 0.01$, $E = 3$, $\alpha = 7.5 \times 10^{-5}$, and a batch size of 16. We adopt the ResNet-8 model [42] for CIFAR-100 dataset with $\eta(t) = 0.1$, $E = 2$, $\alpha = 0.01$, and a batch size of 64. The learning rate of CIFAR-100 is decayed by 0.991 at each communication round.

| Algorithms | FMNIST | | | CIFAR10 | | | CIFAR100 | | |
|---|---|---|---|---|---|---|---|---|---|
| | Acc | Comm (Gbit) | FLOPs (e+11) | Acc | Comm (Gbit) | FLOPs (e+12) | Acc | Comm (Gbit) | FLOPs (e+11) |
| SpaFL | **90.31±0.35** | **1.0208** | 2.7210 | **73.85±2.80** | **2.4956** | **1.5901** | **38.80±1.10** | **0.7674** | **2.8424** |
| FedAvg | 88.78±0.20 | 133.8 | 6.2044 | 61.33±0.15 | 258.36 | 7.4729 | 26.46±0.10 | 26.784 | 19.380 |
| LG-FedAvg | 89.42±0.57 | 1.6 | 6.2044 | 67.43±1.73 | 5.6524 | 7.4729 | 36.67±0.38 | 2.7353 | 19.380 |
| PruneFL | 86.72±0.17 | 70.195 | 5.3443 | 57.19±0.24 | 131.23 | 4.5851 | 22.04±0.12 | 11.477 | 13.555 |
| Sub-FedAvg | 89.29±0.69 | 108.46 | 3.2947 | 70.05±1.88 | 198.52 | 3.5914 | 31.05±1.12 | 25.382 | 14.137 |
| LotteryFL | 89.15±0.62 | 70.195 | 3.0515 | 66.82±0.11 | 204.09 | 3.9700 | 28.90±0.22 | 21.680 | 14.712 |
| FedDST | 84.46±0.14 | 74.461 | 2.8890 | 60.18±0.03 | 139.19 | 3.4373 | 22.26±0.13 | 13.276 | 8.843 |
| FedSpa | 89.30± 0.20 | 55.256 | 5.2510 | 67.63± 0.05 | 129.31 | 4.2978 | 36.32 ±0.03 | 10.203 | 9.275 |
| FedPM | 63.18± 1.74 | 66.554 | **2.1240** | 52.05± 0.06 | 133.19 | 2.7880 | 16.96 ± 0.10 | 13.320 | 5.528 |

Table 1: Performance of SpaFL and other baselines along with their used communication costs (Comm) and computation (FLOPs) resources during whole training.

| Algorithms | Acc | Comm (Gbit) | FLOPs (e+10) | Model Density |
|---|---|---|---|---|
| SpaFL | **89.98±0.5** | **2.1554** | **1.0264** | **4.83%** |
| FedAvg | 88.11±2.7 | 40.366 | 4.1059 | 100% |
| LG-FedAvg | 89.35±0.1 | 6.6113 | 4.1059 | 100% |

Table 2: Performance of SpaFL and other baselines along with their used communication costs (Comm) and computation (FLOPs) resources during whole training with the LSTM model.

## 5.2 BASELINES

We compare SpaFL with seven state of the art baselines that include both dense and sparse FL algorithms. In **FedAVG** [1], every client trains a global dense model and communicates whole model parameters. **LG-FedAvg** [2] is a dense personalized FL algorithm, where clients learn local representations and share only small global layers for communication efficiency. **PruneFL** [6] learns a global sparse model after pruning a dense model at a particular client for initialization. **Sub-FedAvg** [9] is a scheme that learns a personalized sparse model for each client and iteratively performs pruning using validation dataset during training. **LotteryFL** [7] optimizes personalized sparse models by communicating lottery tickets of clients at every round. In **FedDST** [5], clients learn a global sparse model and a global binary mask while communicating only their sparse model parameters and binary masks. **FedSpa** [24] trains personalized sparse models for clients while maintaining fixed model density during training. **FedPM** [30] trains and communicates a binary mask while freezing model parameters. For the sparse FL baselines, the target sparsity is set to 0.5 following the configurations in [5, 6, 7, 9, 24].

## 5.3 MAIN RESULTS

In Table 1 and Fig. 2, we present the averaged accuracy, communication costs, number of FLOPs during training, and convergence rate for each algorithm. We consider all uplink and downlink communications to calculate the communication cost of each algorithm. We also provide the details of the FLOPs measure in the Supplementary document. We average the model densities of SpaFL when a model achieved the best accuracy during training. From these results, we observe that SpaFL outperforms all baselines while using the least amount of communication costs and number of FLOPs except FMNIST dataset. The achieved model densities are 5.36%, 7.57%, and 7.38%, for FMNIST, CIFAR-10, and CIFAR-100, respectively. We also observe that SpaFL uses less resources and performs better than Sub-FedAvg and LotteryFL, which also personalize both model parameters and sparse model structures. Although FedPM reduced uplink communication costs by communicating only binary masks, its downlink cost is the same as FedAvg. In SpaFL, since the clients and the server only exchange thresholds, we can significantly reduce the communication costs compared to baselines that exchange the subset of model parameters such as LG-FedAvg, Sub-FedAvg, LotteryFL, PruneFL, and FedDST. Hence, SpaFL can efficiently improve model performance with small computation and communication costs. In Table 2, we also compare the performance and resource usage of SpaFL with baselines that considered LSTM models in their work. In Fig. 2, we show the convergence rate of each algorithm. We can see that the accuracy of SpaFL decreases and then keeps increasing. The initial accuracy drop is from pruning while global thresholds are not trained enough. As thresholds keep being trained and communicated, clients learn how to prune their model, thereby gradually improving the performance even at high model sparsity.

In Fig. 3, we show the change of model density of SpaFL during training with a different sparsity coefficient $\alpha$. From Fig. 3, we can observe that the model density fluctuates at its low value. This is

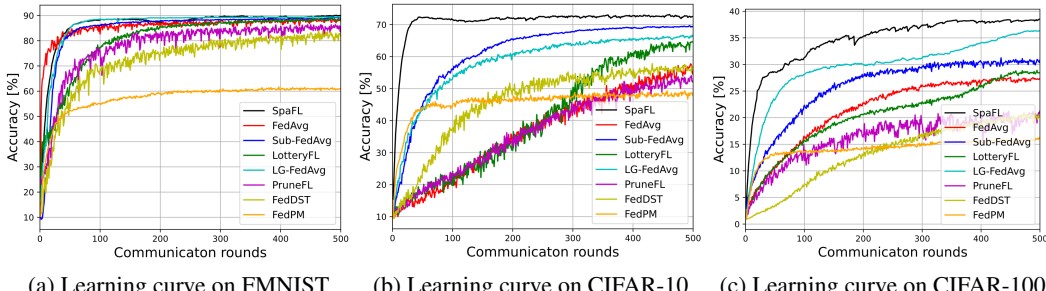

(a) Learning curve on FMNIST     (b) Learning curve on CIFAR-10     (c) Learning curve on CIFAR-100

Figure 2: Learning curves on FMNIST, CIFAR-10, and CIFAR-100

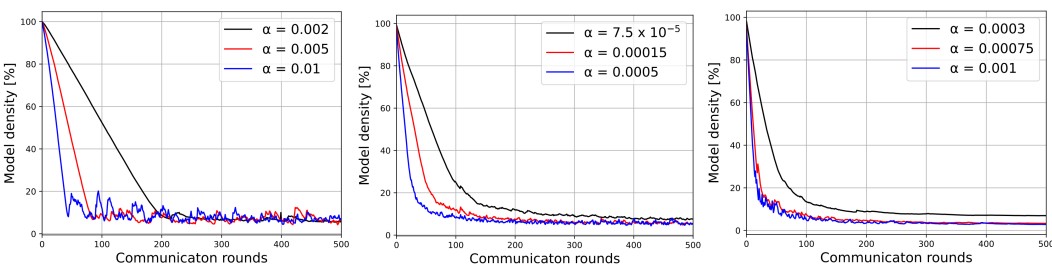

(a) Density change on FMNIST     (b) Density change on CIFAR-10     (c) Density change on CIFAR-100

Figure 3: Model density with different $\alpha$

| Datasets | $\alpha$ | Acc | FLOPs | Model density |
|---|---|---|---|---|
| | 0.0003 | **90.31±0.35** | 2.7210 e+11 | 5.36% |
| FMNIST | 0.0005 | 89.70±0.21 | 0.9024 e+11 | 4.64% |
| | 0.001 | 89.35±0.1 | **0.1344 e+11** | **4.54%** |
| | $7.5 \times 10^{-5}$ | **73.85±2.80** | 1.5901 e+12 | 7.57% |
| CIFAR-10 | 0.00015 | 73.60±2.60 | 0.9416 e+12 | 6.66% |
| | 0.0005 | 73.00±1.45 | **0.4401 e+12** | **5.88%** |
| | 0.0003 | **38.80±1.10** | 2.8424 e+11 | 7.38% |
| CIFAR-100 | 0.00075 | 36.68±0.12 | 1.8556 e+11 | 3.32% |
| | 0.001 | 36.31±0.52 | **1.4628 e+11** | **2.87%** |

Table 3: Performance, communication costs (Comm) and computation (FLOPs) resources of SpaFL with different $\alpha$.

because we are jointly optimizing both model parameters and thresholds. Hence, prematurely pruned parameters will be recovered during training. In Fig. 3 and Table 3, we observe the tradeoff between model performance and computational costs. As $\alpha$ increases, we can quickly enforce sparsity to the model. However, the performance can be low if we set $\alpha$ to be very large. This is because binary masks can change too quickly, thereby making training unstable. Therefore, Fig. 3 also corroborates our theoretical analysis in Theorem 1.

## 6 CONCLUSION

In this paper, we have developed a communication-efficient FL framework SpaFL that allows clients to optimize both personalized model parameters and sparse structures with low computational costs. We have reduced computational overhead by performing pruning through trainable thresholds. To further reduce communication costs, we have communicated only thresholds between clients and a server. We have also presented the parameter update method that can extract parameter importance from global thresholds. Furthermore, we have provided theoretical insights on the convergence of thresholds and experimental results to demonstrate the resource-efficiency of SpaFL.

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

## A   SUPPLEMENTARY DOCUMENT

### A.1   EXPERIMENTS

#### A.1.1   IMPLEMENTATION DETAIL

We run all experiments on NVIDIA A100 GPUs with PyTorch. In Table 4, we provide detailed information of model architectures for each dataset. For the FMNIST dataset, we use the Lenet-5-Caffe model, which is Caffe variant of Lenet-5, and the LSTM model with two LSTM layers. The Lenet model has 430500 of model parameters and 580 of trainable thresholds. The LSTM model has 210944 of model parameters and 2048 of trainable thresholds. For the CIFAR-10 dataset, we use a CNN model of seven layers used in [41]. It has 807366 of model parameters and 1418 of trainable thresholds. The ResNet-8 model [42] is adopted for the CIFAR-100 dataset with 84187 of model parameters and 436 of thresholds. We use a stochastic gradient optimizer with momentum of 0.9. For FMNIST with the Lenet model, we use $\eta(t) = 0.001$, $E = 3$, a batch size of 64, and $\alpha = 0.002$. We set $\eta(t) = 0.01$, $E = 3$, a batch size of 16, and $\alpha = 0.0003$ for the LSTM model. For CIFAR-10, we use $\eta(t) = 0.01$, $E = 3$, a batch size of 16, and $\alpha = 7.5 \times 10^{-5}$. For CIFAR-100, we use $\eta(t) = 0.1$, $E = 2$ decayed by 0.991 at each communication round, a batch size of 64, and $\alpha = 0.01$. All trainable thresholds are initialized to zero. We noticed that too large sparsity coefficient $\alpha$ can dominate the training loss, resulting in masking whole parameters in a certain layer. Following the implementation of [32], if a certain layer's density becomes less than 1%, the corresponding trainable thresholds will be reset to zero to avoid masking whole parameters.

Table 4: Model architectures used in our experiments

|  | FMNIST | CIFAR-10 | CIFAR-100 |
|---|---|---|---|
| **Conv** | (5, 5, out = 20, stride = 1)
Maxpool2d
(5, 5, out = 50, stride = 1)
Maxpool2d | (5, 5, out = 64, stride = 1)
(5, 5, out = 64, stride = 1)
Maxpool2d
(5, 5, out = 128 stride = 1)
(5, 5, out = 128, stride = 1)
Maxpool2d | (3, 3, out = 16, stride = 1)
(3, 3, out = 16, stride = 1)
(3, 3, out = 16, stride = 1)
(3, 3, out = 32, stride = 1)
(3, 3, out = 32, stride = 1)
(3, 3, out = 64, stride = 1)
(3, 3, out = 64, stride = 1) |
| **FC** | (800, 500)
(500, 10) | (512, 128)
(128, 128)
(128, 100) | (64, 100) |

### A.1.2 MORE DETAILS ABOUT BASELINES

We compare SpaFL with seven baselines that include both dense and sparse FL algorithms. In **FedAvg** [1], every client trains a global dense model and communicates whole model parameters. We used the equal weighted average for the model aggregation. **LG-FedAvg** [2] is a dense personalized FL algorithm, where clients learn local representations and share only small global layers for communication efficiency. We communicated the last layer and the last three layers of the FMNIST and CIFAR10/100 models, respectively, for communication efficiency. For the LSTM mdoel, we communicated the last fully-connected layer and weights of the output gate. **PruneFL** [6] learns a global sparse model after pruning a dense model at a particular client for initialization. We randomly chose a client to perform initial pruning, and set the mask readjustment period as 50 communication rounds. **Sub-FedAvg** [9] is a scheme that learns a personalized sparse model for each client. Every client iteratively prunes its model using the validation dataset during training. We set the pruning ratio as 20% for each pruning process. **LotteryFL** [7] optimizes personalized sparse models by communicating lottery tickets of clients at every round. Each client prunes its model using the validation dataset during training and resets unpruned parameters to their initial values. We set the pruning ratio as 20% for each pruning process. In **FedDST** [5], clients learn a global sparse model and a global binary mask while communicating only their sparse model parameters and binary masks. At every predefined communication round, each client performs mask readjustment by pruning and reallocating a certain portion of parameters. We set the readjustment period as 15 and the reallocating ratio as 0.01. **FedSpa** [24] trains personalized sparse models for clients while maintaining fixed model density during training. The initial pruning rate is set to be 0.5 and decayed using cosine annealing. **FedPM** [30] optimizes a binary mask while freezing model parameters. Clients only transmit their arithmetically coded binary masks to the server, and the server broadcasts real-valued probability masks to the clients. We use Adam optimizer with learning rate of $0.1$ as done in [30]. For all the sparse FL baselines, the target sparsity is set to 0.5 following the configurations in [6, 7, 9, 5]. We provide the learning rates of the baselines in the following table.

| Algorithm | FMNIST | CIFAR-10 | CIFAR-100 |
|---|---|---|---|
| FedAvg | $\eta(t) = 0.001$ | $\eta(t) = 0.01$ | $\eta(t) = 0.1$ |
| Sub-FedAvg | $\eta(t) = 0.001$ | $\eta(t) = 0.001$ | $\eta(t) = 0.2$ |
| LG-FedAvg | $\eta(t) = 0.001$ | $\eta(t) = 0.005$ | $\eta(t) = 0.0015$ |
| PruneFL | $\eta(t) = 0.01$ | $\eta(t) = 0.001$ | $\eta(t) = 0.05$ |
| LotteryFL | $\eta(t) = 0.001$ | $\eta(t) = 0.005$ | $\eta(t) = 0.01$ |
| FedDST | $\eta(t) = 0.001$ | $\eta(t) = 0.01$ | $\eta(t) = 0.01$ |
| FedSpa | $\eta(t) = 0.001$ | $\eta(t) = 0.01$ | $\eta(t) = 0.1$ |
| FedPM | $\eta(t) = 0.15$ | $\eta(t) = 0.1$ | $\eta(t) = 0.1$ |

Table 5: learning rates used by the baselines

A.2 PROOF OF THEOREM 1

We next present the detailed proof of Theorem 1. To facilitate the proof, we first restate the update rule of thresholds and used assumptions in the main paper. In SpaFL, thresholds $\boldsymbol{\tau}(t)$ are updated as follows

$$\boldsymbol{\tau}_k(t) = \boldsymbol{\tau}(t) - \eta(t)\boldsymbol{h}_k(\tilde{\boldsymbol{w}}_k^{E-1}(t)) + \alpha\exp(-\boldsymbol{\tau}(t)), \quad ||\boldsymbol{h}_k(\tilde{\boldsymbol{w}}_k^{E-1}(t))||^2 = \sum_{l=1}^{L}\sum_{i=1}^{n_{\text{out}}^l}||h_{k,i}^l(\tilde{\boldsymbol{w}}_k^{E-1}(t))||^2. \tag{16}$$

We then prove Theorem 1 under the following commonly adopted assumption [37].

**Assumption 3.** *(smoothness) $F_k(\cdot)$ is $M$-smooth for $\boldsymbol{\tau}$ and client $k$, $\forall k$*

$$F_k(\boldsymbol{w}, \boldsymbol{\tau}') \leq F_k(\boldsymbol{w}, \boldsymbol{\tau}) + \langle\nabla_{\boldsymbol{\tau}}F_k(\boldsymbol{w}, \boldsymbol{\tau}), \boldsymbol{\tau}' - \boldsymbol{\tau}\rangle + \frac{M}{2}||\boldsymbol{\tau}' - \boldsymbol{\tau}||^2, \ \forall \boldsymbol{\tau}. \tag{17}$$

**Assumption 4.** *(Unbiased stochastic gradient) The stochastic gradient $\boldsymbol{h}_k$ is an unbiased estimator of the gradient $\nabla_{\boldsymbol{\tau}}F_k$, respectively, for client $k, \forall k$, such that*

$$\mathbb{E}\boldsymbol{h}_k(\boldsymbol{w}_k) = \nabla_{\boldsymbol{\tau}}F_k(\boldsymbol{w}_k, \boldsymbol{\tau}). \tag{18}$$

We first consider the case in which global thresholds converge. We have the following update rule for global thresholds as

$$\boldsymbol{\tau}(t+1) = \frac{1}{K}\sum_{k\in\mathcal{S}_t}\boldsymbol{\tau}_k(t) = \boldsymbol{\tau}(t) - \frac{1}{K}\eta(t)\sum_{k\in\mathcal{S}_t}\boldsymbol{h}_k(\tilde{\boldsymbol{w}}_k^{E-1}(t)) + \alpha\eta(t)\exp(-\boldsymbol{\tau}(t)). \tag{19}$$

We take the expectation over the randomness in client scheduling and stochastic gradients as follows

$$\mathbb{E}\boldsymbol{\tau}(t+1) = \boldsymbol{\tau}(t) - \frac{\eta(t)}{K}\mathbb{E}\sum_{k\in\mathcal{S}_t}\boldsymbol{h}_k(\tilde{\boldsymbol{w}}_k^{E-1}(t)) + \alpha\eta(t)\exp(-\boldsymbol{\tau}(t)).$$

$$= \boldsymbol{\tau}(t) - \frac{\eta(t)}{N}\mathbb{E}\sum_{k=1}^{N}\nabla_{\boldsymbol{\tau}}F_k(\tilde{\boldsymbol{w}}_k^{E-1}(t), \boldsymbol{\tau}(t)) + \alpha\eta(t)\exp(-\boldsymbol{\tau}(t)). \tag{20}$$

Hence, clearly $\boldsymbol{\tau}$ will eventually converge if $\frac{1}{N}\mathbb{E}||\sum_{k=1}^{N}\nabla_{\boldsymbol{\tau}}F_k(\tilde{\boldsymbol{w}}_k^{E-1}(t), \boldsymbol{\tau}(t))||^2$ converges. We next show that this conditional statement holds in our SpaFL framework.

From the $M$-smoothness of the loss function in Assumption 3, we have

$$F_k(\tilde{\boldsymbol{w}}_k^{E-1}(t), \boldsymbol{\tau}_k(t)) \leq F_k(\tilde{\boldsymbol{w}}_k^{E-1}(t), \boldsymbol{\tau}(t)) + \langle\nabla_{\boldsymbol{\tau}}F_k(\tilde{\boldsymbol{w}}_k^{E-1}(t), \boldsymbol{\tau}(t)), \boldsymbol{\tau}_k(t) - \boldsymbol{\tau}(t)\rangle + \frac{M}{2}||\boldsymbol{\tau}_k(t) - \boldsymbol{\tau}(t)||^2 \tag{21}$$

To facilitate the analysis, we first derive $\boldsymbol{\tau}_k(t) - \boldsymbol{\tau}(t)$ as below

$$\boldsymbol{\tau}_k(t) - \boldsymbol{\tau}(t) = -\eta(t)\boldsymbol{h}_k(\tilde{\boldsymbol{w}}_k^{E-1}(t)) + \alpha\eta(t)\exp(-\boldsymbol{\tau}(t)). \tag{22}$$

Then, we can change (21) as follows

$$F_k(\tilde{\boldsymbol{w}}_k^{E-1}(t), \boldsymbol{\tau}_k(t)) \leq F_k(\tilde{\boldsymbol{w}}_k^{E-1}(t), \boldsymbol{\tau}(t)) + \langle\nabla_{\boldsymbol{\tau}}F_k(\tilde{\boldsymbol{w}}_k^{E-1}(t), \boldsymbol{\tau}(t)), -\eta(t)\boldsymbol{h}_k(\tilde{\boldsymbol{w}}_k^{E-1}(t))\rangle$$
$$+ \langle\nabla_{\boldsymbol{\tau}}F(\tilde{\boldsymbol{w}}_k^{E-1}(t), \boldsymbol{\tau}(t)), \alpha\eta(t)\exp(-\boldsymbol{\tau}(t))\rangle$$
$$+ \frac{M\eta(t)^2}{2}||\boldsymbol{h}_k(\tilde{\boldsymbol{w}}_k^{E-1}(t)) - \alpha\eta(t)\exp(-\boldsymbol{\tau}(t))||^2. \tag{23}$$

We next take the expectation to the above inequality and use Assumption 4 as below

$$
\begin{aligned}
\mathbb{E}F_k(\tilde{\boldsymbol{w}}_k^{E-1}(t), \boldsymbol{\tau}_k(t)) \leq{}& \mathbb{E}F_k(\tilde{\boldsymbol{w}}_k^{E-1}(t), \boldsymbol{\tau}(t)) + \langle \nabla_{\boldsymbol{\tau}}F_k(\tilde{\boldsymbol{w}}_k^{E-1}(t), \boldsymbol{\tau}(t)), -\eta(t)\nabla_{\boldsymbol{\tau}}F_k(\tilde{\boldsymbol{w}}_k^{E-1}(t), \boldsymbol{\tau}(t))\rangle \\
&+ \langle \nabla_{\boldsymbol{\tau}}F_k(\tilde{\boldsymbol{w}}_k^{E-1}(t), \boldsymbol{\tau}(t)), \alpha\eta(t)\exp(-\boldsymbol{\tau}(t))\rangle \\
&+ \frac{M\eta(t)^2}{2}\mathbb{E}||\boldsymbol{h}_k(\tilde{\boldsymbol{w}}_k^{E-1}(t)) - \alpha\exp(-\boldsymbol{\tau}(t))||^2 \\
={}& \mathbb{E}F_k(\tilde{\boldsymbol{w}}_k^{E-1}(t), \boldsymbol{\tau}(t)) - \eta(t)||\nabla_{\boldsymbol{\tau}}F_k(\tilde{\boldsymbol{w}}_k^{E-1}(t), \boldsymbol{\tau}(t))||^2 \\
&+ \underbrace{\alpha\eta(t)(1 - M\eta(t))\langle\nabla_{\boldsymbol{\tau}}F_k(\tilde{\boldsymbol{w}}_k^{E-1}(t), \tau(t)), \exp(-\boldsymbol{\tau}(t))\rangle}_{A} \\
&+ \underbrace{\frac{M\eta(t)^2}{2}\mathbb{E}||\boldsymbol{h}_k(\tilde{\boldsymbol{w}}_k^{E-1}(t))||^2}_{B} + \frac{M\alpha^2\eta(t)^2}{2}||\exp(-\boldsymbol{\tau}(t))||^2. \quad (24)
\end{aligned}
$$

We first bound $A$ using $\langle a, b\rangle \leq \frac{||a||^2 + ||b||^2}{2}$ as below

$$
A \leq \frac{\alpha\eta(t)(1 - M\eta(t))}{2}\left[||\nabla_{\boldsymbol{\tau}}F_k(\tilde{\boldsymbol{w}}_k^{E-1}(t), \boldsymbol{\tau}(t))||^2 + ||\exp(-\boldsymbol{\tau}(t))||^2\right]. \quad (25)
$$

We now further bound $B$ using (16) as

$$
\begin{aligned}
B ={}& \frac{M\eta(t)^2}{2}\mathbb{E}\sum_{l=1}^{L}\sum_{i=1}^{n_{\text{out}}^l}||\sum_{j=1}^{n_{\text{in}}^l}\{\boldsymbol{g}_k(\tilde{\boldsymbol{w}}_k^{E-1}(t))\}_{ij}^l w_{k,ij}^{E-1,l}(t)||^2 \\
\leq{}& \frac{M\eta(t)^2}{2}\mathbb{E}\sum_{l=1}^{L}\sum_{i=1}^{n_{\text{out}}^l}n_{\text{in}}^l\sum_{j=1}^{n_{\text{in}}^l}||\{\boldsymbol{g}_k(\tilde{\boldsymbol{w}}_k^{E-1}(t))\}_{ij}^l w_{k,ij}^{E-1,l}(t)||^2 \\
\leq{}& \frac{M\eta(t)^2 n_{\text{in}}^{\max}}{2}\mathbb{E}\sum_{l=1}^{L}\sum_{i=1}^{n_{\text{out}}^l}\sum_{j=1}^{n_{\text{in}}^l}||\{\boldsymbol{g}_k(\tilde{\boldsymbol{w}}_k^{E-1}(t))\}_{ij}^l w_{k,ij}^{E-1,l}(t)||^2 \\
\overset{(a)}{\leq}{}& \frac{M\eta(t)^2 n_{\text{in}}^{\max}}{2}\mathbb{E}\sum_{l=1}^{L}\sum_{i=1}^{n_{\text{out}}^l}\sum_{j=1}^{n_{\text{in}}^l}||\{\boldsymbol{g}_k(\tilde{\boldsymbol{w}}_k^{E-1}(t))\}_{ij}^l||^2 \\
={}& \frac{M\eta(t)^2 n_{\text{in}}^{\max}}{2}\mathbb{E}||\boldsymbol{g}_k(\tilde{\boldsymbol{w}}_k^{E-1}(t))||^2 \leq M^2\eta(t)^2 n_{\text{in}}^{\max}F_k(\tilde{\boldsymbol{w}}_k^{E-1}, \boldsymbol{\tau}(t)), \quad (26)
\end{aligned}
$$

where $n_{\text{in}}^{\max}$ is the largest number of parameters connected to a neuron or filter in a given model, $(a)$ is from $|w| \leq 1$ in Section 3.2.1, and the last inequality is from the $M$-smoothness of $F_k$. By combining $A$ and $B$ with taking expectation, we have

$$
\begin{aligned}
\mathbb{E}F_k(\tilde{\boldsymbol{w}}_k^{E-1}(t), \boldsymbol{\tau}_k(t)) \leq{}& \mathbb{E}F_k(\tilde{\boldsymbol{w}}_k^{E-1}(t), \tau(t)) - \eta(t)\left\{1 - \frac{\alpha(1 - M\eta(t))}{2}\right\}||\nabla_{\boldsymbol{\tau}}F_k(\tilde{\boldsymbol{w}}_k^{E-1}(t), \boldsymbol{\tau}(t))||^2 \\
&+ \frac{\alpha\eta(t)(1 - M\eta(t)(1 - \alpha))}{2}||\exp(-\boldsymbol{\tau}(t))||^2 + M^2\eta(t)^2 n_{\text{in}}^{\max}\mathbb{E}F_k(\tilde{\boldsymbol{w}}_k^{E-1}, \boldsymbol{\tau}(t)) \quad (27)
\end{aligned}
$$

By arranging the above inequality, we have

$$
\begin{aligned}
||\nabla_{\boldsymbol{\tau}}F(\tilde{\boldsymbol{w}}_k^{E-1}(t), \boldsymbol{\tau}(t))||^2 \leq{}& \frac{1}{\gamma(t)}\left[\mathbb{E}\underbrace{F_k(\tilde{\boldsymbol{w}}_k^{E-1}(t), \tau(t)) - \mathbb{E}F_k(\tilde{\boldsymbol{w}}_k^{E-1}(t), \tau_k(t))}_{(A)}\right] \\
&+ \frac{\alpha\eta(t)}{\gamma(t)}\{1 - M\eta(t)(1 - \alpha)\}||\exp(-\boldsymbol{\tau}(t))||^2 + \frac{M^2\eta(t)^2 n_{\text{in}}^{\max}}{\gamma(t)}\mathbb{E}F_k(\tilde{\boldsymbol{w}}_k^{E-1}, \boldsymbol{\tau}(t)), \quad (28)
\end{aligned}
$$

where $\gamma(t) = \eta(t)(1 - \frac{\alpha(1-M\eta(t))}{2})$. We now further bound $(A)$ in (28). From Assumption 3, we have the following

$$(A) \leq \langle \nabla_{\boldsymbol{\tau}} F_k(\tilde{\boldsymbol{w}}_k^{E-1}(t), \boldsymbol{\tau}(t)), \boldsymbol{\tau}(t) - \boldsymbol{\tau}_k(t) \rangle + \frac{1}{2M} ||\nabla_{\boldsymbol{\tau}} F_k(\tilde{\boldsymbol{w}}_k^{E-1}(t), \boldsymbol{\tau}(t)) - \nabla_{\boldsymbol{\tau}_k} F_k(\tilde{\boldsymbol{w}}_k^{E-1}(t), \boldsymbol{\tau}_k(t))||^2$$

$$\leq \frac{\gamma(t)}{2} ||\nabla_{\boldsymbol{\tau}} F_k(\tilde{\boldsymbol{w}}_k^{E-1}(t), \boldsymbol{\tau}(t))||^2 + \frac{1}{2\gamma(t)} ||\boldsymbol{\tau}(t) - \boldsymbol{\tau}_k(t)||^2$$

$$+ \frac{1}{2M} ||\nabla_{\boldsymbol{\tau}} F_k(\tilde{\boldsymbol{w}}_k^{E-1}(t), \boldsymbol{\tau}(t)) - \nabla_{\boldsymbol{\tau}_k} F_k(\tilde{\boldsymbol{w}}_k^{E-1}(t), \boldsymbol{\tau}_k(t))||^2. \tag{29}$$

Based on (29), we can bound (28) as below

$$||\nabla_{\boldsymbol{\tau}} F(\tilde{\boldsymbol{w}}_k^{E-1}(t), \boldsymbol{\tau}(t))||^2 \leq \frac{1}{M\gamma(t)} \mathbb{E} ||F_k(\nabla_{\boldsymbol{\tau}(t)} \tilde{\boldsymbol{w}}_k^{E-1}(t), \tau(t)) - \nabla_{\boldsymbol{\tau}_k(t)} F_k(\tilde{\boldsymbol{w}}_k^{E-1}(t), \tau_k(t))||^2$$

$$+ \frac{2\alpha\eta(t)}{\gamma(t)} \{1 - M\eta(t)(1-\alpha)\} ||\exp(-\boldsymbol{\tau}(t))||^2 + \frac{2M^2\eta(t)^2 n_{\text{in}}^{\max}}{\gamma(t)} \mathbb{E} F_k(\tilde{\boldsymbol{w}}_k^{E-1}, \boldsymbol{\tau}(t))$$

$$+ \frac{||\boldsymbol{\tau}(t) - \boldsymbol{\tau}_k(t)||^2}{\gamma(t)^2}. \tag{30}$$

From (30), we can bound the averaged aggregated gradients with respect to thresholds as below

$$\frac{1}{N} \mathbb{E} ||\sum_{k=1}^{N} \nabla_{\boldsymbol{\tau}} F(\tilde{\boldsymbol{w}}_k^{E-1}(t), \boldsymbol{\tau}(t))||^2 \leq \frac{1}{N} \sum_{k=1}^{N} \mathbb{E} ||\nabla_{\boldsymbol{\tau}} F(\tilde{\boldsymbol{w}}_k^{E-1}(t), \boldsymbol{\tau}(t))||^2$$

$$\leq \frac{1}{NM\gamma(t)} \left( \sum_{k=1}^{N} \mathbb{E} ||F_k(\nabla_{\boldsymbol{\tau}(t)} \tilde{\boldsymbol{w}}_k^{E-1}(t), \tau(t)) - \nabla_{\boldsymbol{\tau}_k(t)} F_k(\tilde{\boldsymbol{w}}_k^{E-1}(t), \tau_k(t))||^2 \right)$$

$$+ \frac{2\alpha\eta(t)}{\gamma(t)} \{1 - M\eta(t)(1-\alpha)\} ||\exp(-\boldsymbol{\tau}(t))||^2$$

$$+ \frac{1}{N} \sum_{k=1}^{N} \frac{2M^2\eta(t)^2 n_{\text{in}}^{\max}}{\gamma(t)} \mathbb{E} F_k(\tilde{\boldsymbol{w}}_k^{E-1}, \boldsymbol{\tau}(t)) + \frac{1}{N} \sum_{k=1}^{N} \frac{\mathbb{E} ||\boldsymbol{\tau}(t) - \boldsymbol{\tau}_k(t)||^2}{\gamma(t)^2}. \tag{31}$$

By summing the above inequality from $t = 0$ to $t = T - 1$, we can obtain the result of Theorem 1.

### A.2.1 BOUNDING THE LOSS FUNCTION

In Theorem 1, we have the loss function $F_k$, which is not assumed to be bounded in our analysis. Although loss functions can generally be a large value, we constrained model parameters $\boldsymbol{w}$ and thresholds $\boldsymbol{\tau}$ to be within $[-1, 1]$ and $[0, 1]$, respectively. Therefore, if input-output pairs $x, y$ are drawn from some bounded domain, the loss function $F_k$ can be obviously bounded by a finite real value. However, since inputs can be sampled from unbounded domain, we provide Corollary below to mitigate this issue.

**Corollary 1.** *Assume that there exists $G \geq 0$ such that $\mathbb{E} ||g_k(\boldsymbol{w}_k)||^2 \leq G^2, \forall k$. For $\gamma(t) = \eta(t)(1 - \frac{\alpha(1-M\eta(t))}{2})$ and the largest number of parameters connected to a neuron or filter $n_{in}^{max} > 0$ in a given model, we have*

$$\frac{1}{NT} \sum_{t=0}^{T-1} \mathbb{E} ||\sum_{k=1}^{N} \nabla_{\boldsymbol{\tau}} F_k(\tilde{\boldsymbol{w}}_k^{E-1}(t), \boldsymbol{\tau}(t))||^2 \leq \sum_{t=0}^{T-1} \sum_{k=1}^{N} \frac{\mathbb{E} ||\nabla_{\boldsymbol{\tau}} F_k(\tilde{\boldsymbol{w}}_k^{E-1}(t), \boldsymbol{\tau}(t)) - \nabla_{\boldsymbol{\tau}_k} F_k(\tilde{\boldsymbol{w}}_k^{E-1}(t), \boldsymbol{\tau}_k(t))||^2}{MNT\gamma(t)}$$

$$+ \sum_{t=0}^{T-1} \frac{2\alpha\eta(t)}{T\gamma(t)} \{1 - M\eta(t)(1-\alpha)\} ||\exp(-\boldsymbol{\tau}(t))||^2$$

$$+ \sum_{t=0}^{T-1} \frac{M^2\eta(t)^2 n_{in}^{max}}{2T\gamma(t)} G^2 + \sum_{t=0}^{T-1} \sum_{k=1}^{N} \frac{\mathbb{E} ||\boldsymbol{\tau}(t) - \boldsymbol{\tau}_k(t)||^2}{NT\gamma(t)}. \tag{32}$$

*Proof.* We start from bounding the term $B$ in eq. (24) as follows

$$
\begin{aligned}
B &= \frac{M\eta(t)^2}{2}\mathbb{E}\sum_{l=1}^{L}\sum_{i=1}^{n_{\text{out}}^l}||\sum_{j=1}^{n_{\text{in}}^l}\{\boldsymbol{g}_k(\tilde{\boldsymbol{w}}_k^{E-1}(t))\}_{ij}^l w_{k,ij}^{E-1,l}(t)||^2 \\
&\leq \frac{M\eta(t)^2}{2}\mathbb{E}\sum_{l=1}^{L}\sum_{i=1}^{n_{\text{out}}^l} n_{\text{in}}^l \sum_{j=1}^{n_{\text{in}}^l}||\{\boldsymbol{g}_k(\tilde{\boldsymbol{w}}_k^{E-1}(t))\}_{ij}^l w_{k,ij}^{E-1,l}(t)||^2 \\
&\leq \frac{M\eta(t)^2 n_{\text{in}}^{\max}}{2}\mathbb{E}\sum_{l=1}^{L}\sum_{i=1}^{n_{\text{out}}^l}\sum_{j=1}^{n_{\text{in}}^l}||\{\boldsymbol{g}_k(\tilde{\boldsymbol{w}}_k^{E-1}(t))\}_{ij}^l w_{k,ij}^{E-1,l}(t)||^2 \\
&\overset{(a)}{\leq} \frac{M\eta(t)^2 n_{\text{in}}^{\max}}{2}\mathbb{E}\sum_{l=1}^{L}\sum_{i=1}^{n_{\text{out}}^l}\sum_{j=1}^{n_{\text{in}}^l}||\{\boldsymbol{g}_k(\tilde{\boldsymbol{w}}_k^{E-1}(t))\}_{ij}^l||^2 \\
&= \frac{M\eta(t)^2 n_{\text{in}}^{\max}}{2}\mathbb{E}||\boldsymbol{g}_k(\tilde{\boldsymbol{w}}_k^{E-1}(t))||^2 \leq \frac{M\eta(t)^2 n_{\text{in}}^{\max}}{2}G^2. \quad (33)
\end{aligned}
$$

By following eq. (30) and (31), we can derive the convergence bound.

$\square$

### A.3 COMMUNICATION COSTS MEASURE

We calculate the communication cost of SpaFL considering both uplink and downlink communications. At each round $t$, sampled clients transmit their updated thresholds to the server. Hence, the uplink communication costs can be given by

$$
\text{Comm}_{\text{Up}} = K \times \boldsymbol{\tau}_{\text{num}} \times 32 \text{ [bits]}, \quad (34)
$$

where $\boldsymbol{\tau}_{\text{num}}$ is the number of thresholds of a given model. In downlink, the server broadcasts the updated global threshold to every client. Hence, the downlink communication costs can be given as below

$$
\text{Comm}_{\text{down}} = N \times \boldsymbol{\tau}_{\text{num}} \times 32 \text{ [bits]}. \quad (35)
$$

Therefore, total communication costs can be given by $T \times (\text{Comm}_{\text{Up}} + \text{Comm}_{\text{down}})$.

### A.4 FLOPS MEASURE

We calculate the number of FLOPs during training using the framework introduced in [43]. We consider a convolutional layer with an input tensor $X \in \mathbb{R}^{N \times C \times X \times Y}$, parameter tensor $W \in \mathbb{R}^{F \times C \times R \times S}$, and output tensor $O \in \mathbb{R}^{N \times F \times H \times W}$. Here, the input tensor $X$ consists of $N$ number of samples, each of which has $X \times Y$ dimension. The parameter tensor $W$ has $F$ filters of $C$ channels with kernel size $R \times S$. The output tensor $O$ will have $F$ output channels with dimension $H \times W$ for $N$ samples. During forward propagation, a filter in $W$ performs convolution operation with the input tensor $X$ to produce a single value in the output tensor $O$. Hence, we can approximate the number of FLOPs as $N \times (C \times R \times S) \times F \times H \times W$. Since we use a sparse model during forward propagation, the number of FLOPs can be reduced to $\rho \times N \times (C \times R \times S) \times F \times H \times W$, where $\rho = \frac{||\boldsymbol{p}||_0}{||W||_0}$ is the density of the parameter matrix $W$. For the backpropagation, we calculate input gradient and parameter gradient. We can calculate the number of FLOPs for the input gradient by convolving the output gradient with the parameter matrix $W$. Hence, it can approximated to $N \times (F \times R \times S) \times C \times X \times Y$. Since we used a sparse model during the forward propagation, this can be reduced to $\rho \times N \times (F \times R \times S) \times C \times X \times Y$. For the parameter gradient, we can calculate it by convolving the input activation with the output gradient. Therefore, this can be approximately given by $(N \times X \times Y) \times F \times C \times R \times S$. Here, we did not block the gradients of input activations. Thus, the FLOPs during the backward propagation can be given by $(1+\rho)(N \times F \times R \times S \times C \times X \times Y)$. Hence, the computational overhead of backpropagation is approximately $(1+\rho)$ times that of the forward propagation.

For a fully connected layer with input tensor $X \in \mathbb{R}^{N \times X}$ and parameter tensor $W \in \mathbb{R}^{X \times Y}$, the input tensor $X$ is multiplied with $W$ during the forward propagation. Hence, with the density of $W$, we can calculate the number of FLOPs for the forward propagation as $\rho \times N \times X \times Y$. In backpropagation, we can obtain the gradient of loss by calculating the input gradient and parameter gradient in a similar manner with convolutional layers. Hence, the number of FLOPs during backpropagation can by given by $(1 + \rho) \times N \times X \times Y$.

For an LSTM layer with input size $I$ and hidden size $D$, we can derive the number of FLOPs of the forward propagation as $4(I \times D \times \rho_1 + D \times D \times \rho_2)$, where $\rho_1$ and $\rho_2$ corresponds to the density of the corresponding weight matrices [44]. Note that an LSTM layer usually has three gates and one memory cell. We approximated the number of FLOPs of backpropagation as $(1 + \frac{\rho_1 + \rho_2}{2}) \times (I \times D \times \rho_1 + D \times D \times \rho_2)$ as done in linear and convolutional layers previously.

The number of FLOPs for model parameter update will be the same as the number of model parameters $d$. For, thresholds update, it can be given by $||\boldsymbol{\tau}||_0 + d$ because we perform multiplication and addition using the connected parameters of each threshold. We also consider the number of FLOPs to perform line 7 in Algorithm 1 for updating the local models from global thresholds. Every client first has to decide update directions by doing summation of connected parameters at each neuron/filter (sum operation). Then, they update their local models using the received global thresholds (sum and multiply operations). This corresponds to $1.5 \times d$ FLOPs, where $d$ is the number of model parameters. Then, the total number of FLOPs during one local epoch at round $t$ can be approximately given by

$$\text{FLOP}(t) = \sum_{l=1}^{L} (2 + \rho_l(t)) N \times (C_l \times R_l \times S_l) \times F_l \times H_l \times W_l \times \mathbb{1}\{\text{layer } l == \text{conv}\}$$
$$+ (2 + \rho_l(t)) \times N \times X_l \times Y_l \times \mathbb{1}\{\text{layer } l == \text{fc}\} + 3.5d + ||\boldsymbol{\tau}||_0. \tag{36}$$

## A.5 Change of Model Sparsity across clients with Different $\alpha$ on FMNIST and CIFAR-10/100

Here, we present the distribution of model sparsity across clients with different $\alpha$ on FMNIST and CIFAR-100 datasets.

### A.5.1 Change of Model Sparsity across clients with Different $\alpha$ on FMNIST

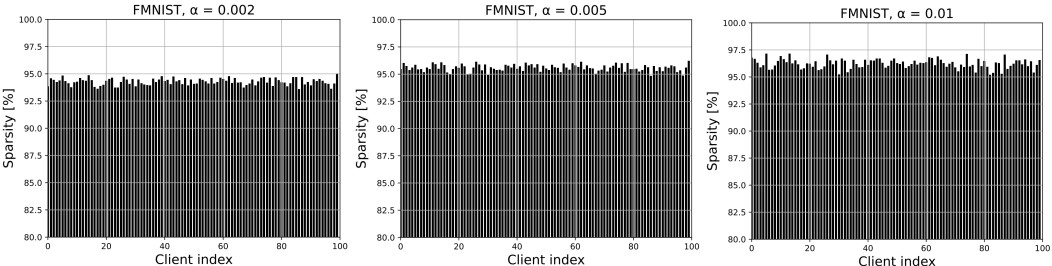

Figure 4: Model sparsity across clients with different $\alpha$

### A.5.2 Change of Model Sparsity across clients with Different $\alpha$ on CIFAR-10

### A.5.3 Change of Model Sparsity across clients with Different $\alpha$ on CIFAR-100

From Figs. 4, 5 and 6, we can observe that every model of clients becomes sparser as $\alpha$ increases clients. We can also see that clients show different model sparsity due to the heterogeneity in data distribution.

### A.6 SpaFL with structured sparsity

SpaFL can be readily extended to perform structured pruning. Specifically, we can prune entire parameters connected to a filter/neuron. In SpaFL, we defined a threshold for each filter/neuron

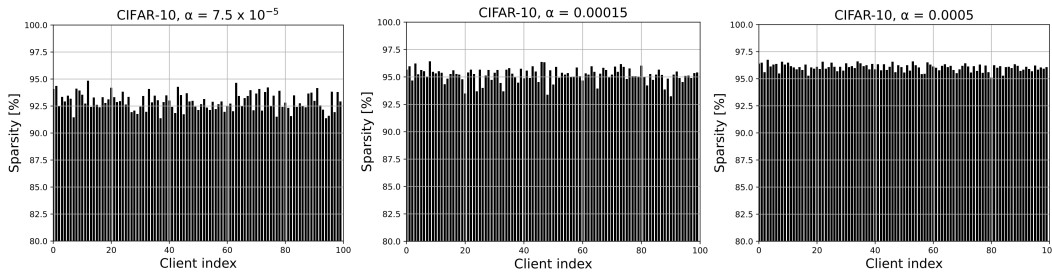

Figure 5: Model sparsity across clients with different $\alpha$

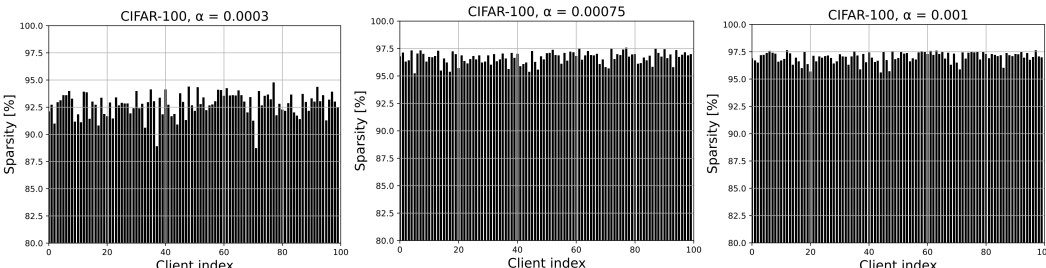

Figure 6: Model sparsity across clients with different $\alpha$

and pruned each connected parameter if its magnitude is smaller than that of the threshold. This approach leads to unstructured sparsity. We can endow SpaFL with structured sparsity by calculating the average magnitude of all parameters connected to their filter/neuron and pruning all of them if their average magnitude is smaller than their threshold. Hence, SpaFL can be extended to do filter/neuron-wise pruning. We provide the performance of SpaFL with structured sparsity below.

| Algorithm | FMNIST | CIFAR-10 | CIFAR-100 |
|---|---|---|---|
| Structured SpaFL | **90.31±0.34** | **73.85±2.80** | **38.80±1.10** |
| Sub-FedAvg | 89.29±0.69 | 70.05±1.88 | 31.05±1.12 |
| LotteryFL | 89.15±0.62 | 66.82±0.11 | 28.90±0.22 |

Table 6: Performance of SpaFL with structured sparsity and baselines with unstructured sparsity.

The achieved model densities are 50.6%, 50%, and 49.3% for FMNIST, CIFAR-10, and CIFAR-100, respectively. The target density of the baselines is set to be 50%. Since it is well known that filter/neuron-wise pruning is less flexible than unstructured pruning, the results show accuracy loss compared to the original SpaFL. However, we can see that the structured SpaFL still outperforms unstructured baselines at the same model density 50%.

## A.7 CHOICE OF SPARSITY REGULARIZER

In this subsection, we investigate the impact of different sparsity regulaizer $R(t)$ for updating thresholds used in (4). We chose $R(t) = \sum_{l=1}^{L} \sum_{i=1}^{n_{\text{out}}^l} \exp(-\tau_i)$ because an exponential function goes to zero asymptotically as thresholds increase. This is reasonable because it can penalize low thresholds without making them become extremely large. We can also use a similar regularizer that can give penalty for low thresholds without encouraging them to become a too large value. For example, a regularizer $R'(t) = \frac{1}{||\boldsymbol{\tau}(t)+0.1}$ satisfies such property. Below, we provide an empirical comparison between SpaFL and the baseline that uses $R'(t)$ in Table 7.

For the regularizer $R(t)$, the achieved model densities are 5.36%,7.57% and 7.38% for FMNIST, CIFAR-10, and CIFAR-100, respectively. Meanwhile, with $R'(t)$, the model densities are 4.85%, 5.60%, and 8.01% for FMNIST, CIFAR-10, and CIFAR-100, respectively. We can see that both regularizers work well with SpaFL. Hence, SpaFL is compatible with any regularizers that can penalize the magnitude of thresholds asymptotically.

| Algorithm | FMNIST | CIFAR-10 | CIFAR-100 |
|---|---|---|---|
| SpaFL | **90.31±0.34** | **73.85±2.80** | **38.80±1.10** |
| Baseline | 89.82 ± 0.25 | 73.62 ± 3.42 | 38.73 ± 0.53 |

Table 7: Comparison between two sparsity regularizaers.

## A.8 IMPACT OF EXTRACTING PARAMETER IMPORTANCE

The main purpose of Section 3.2.4 is to update the model parameters by extracting the aggregated importance from global thresholds. By investigating the difference between two consecutive global thresholds and parameters, clients can deduce which parameters are globally important. Although clients perform their training using the received global thresholds as described in Section 3.2.2, this additional update provides meaningful performance gains. We provide an empirical comparison between SpaFL and the baseline that does not use the update in 3.2.4 in Table. 8.

| Algorithm | FMNIST | CIFAR-10 | CIFAR-100 |
|---|---|---|---|
| SpaFL | **90.31±0.34** | **73.85±2.80** | **38.80±1.10** |
| Baseline | 90.02±0.30 | 72.92±2.15 | 38.06±0.80 |

Table 8: Impact of extracting parameter importance from global thresholds

From Table. 8, we can see that the update in Section 3.2.4. can provide a clear improvement compared to the baseline by utilizing the aggregated parameter importance from global thresholds.

## A.9 SPAFL WITH UNAVAILABLE CLIENTS

Clients may not always be able to receive global thresholds due to their constrained resources. To demonstrate that SpaFL is also compatible with intermittent availability, we did comparison between SpaFL and a baseline where only a small subset of clients can receive the server update. This baseline can capture the unavailability of some clients who cannot participate in the training at a given time. Below, we provide an empirical comparison between SpaFL and the baseline with the scheduling size of 10.

| Algorithm | FMNIST | CIFAR-10 | CIFAR-100 |
|---|---|---|---|
| SpaFL | **90.31±0.34** | **73.85±2.80** | **38.80±1.10** |
| Baseline | 90.24±0.08 | 73.45±3.19 | 38.65±0.43 |

Table 9: Comparison between SpaFL and the scenario with unavailable clients

For SpaFL, the communications costs are 1.0208 Gbit, 2.4956 Gbit, and 0.7674 Gbit, for FMNIST, CIFAR-10, and CIFAR-100, respectively. For the baseline, the communications costs are 0.1856 Gbit, 0.4357 Gbit, and 0.1395 Gbit, for FMNIST, CIFAR-10, and CIFAR-100, respectively. We can see that the performance loss of the baseline is very small. Moreover, the baseline used only 20% of the communication costs used by SpaFL. Therefore, this result demonstrates the compatibility of SpaFL with a scenario where only a subset of clients is available and receives the server update.

## A.10 SPAFL WITH RESNET-18

We train ResNet-18 on CIFAR-100 distributed over $N = 100$ clients in a non-iid fashion following Dirichlet $(0.1)$ for $1000$ rounds. The initial learning rate was $0.01$ and decayed to $0.0001$ by multiplying $0.993$ at each round. We set $\alpha = 0.0005, E = 2, K = 10$, and the bath size of $64$.

| Algorithm | Acc | FLOP(e+13) | Comm (Gbit) |
|---|---|---|---|
| SpaFL | **44.35±0.39** | **4.5288** | **8.448** |
| Baseline | 36.50±0.42 | 16.626 | 3570.9 |

Table 10: SpaFL with ResNet-18 on CIFAR-100.

From Table 10, we can see that SpaFL still works well using significantly less computing and communication resources compared to FedAvg.

