# OpenReview forum: "SpaFL: Communication-Efficient Federated Learning with Sparse Models and Low Computational Overhead"
_ICLR.cc/2024/Conference — Submitted to ICLR 2024_

### Official Review · Reviewer_Kuri · 2023-10-28

**Soundness:** 3 good
**Presentation:** 2 fair
**Contribution:** 3 good
**Rating:** 6
**Confidence:** 5

**Summary:**

The large communication overhead of FL is one of the main challenges. This work proposes SpaFL to optimize both personalized model parameters and sparse model structures. SpaFL defines a trainable threshold for each neuron/filter for pruning. Both model parameters and thresholds are jointly optimized, thus those prematurely pruned parameters during training can be recovered. Only thresholds are communicated between a server and clients instead of parameters, thereby enabling the clients to learn how to prune and reducing communication costs. Global thresholds are used to update model parameters by extracting aggregated parameter importance.

**Strengths:**

1. Only communicating thresholds is novel, and reducing communication costs of both up-link and down-link a lot.
2. Equation (8) provides a good connection between the importance and the thresholds.
3. There is convergence analysis of the SpaFL.
4. Experiment results show significant improvements of SpaFL.

**Weaknesses:**

1. Section 3.2.2 needs to be written more clear. During local training with e < E, does the thresholds not be updated? Equation (5) and (6) only work for the e=E? And how the equation (6) is derived?
2. Comparing equation (10) and (11), authors concludes the relationship between the gradient direction and the $\Delta \tau$ when $w >0$ or $w < 0$. However, Equation (9) is about the global $\tau$, while equation (10) is talking about local $\tau$. Could authos explain this in more details?
3. Experiment settings are not clear enough. The training dataset is split using Dirichlet samplg. For personalized FL, how are test datasets split and how the models are tested? Why all methods use the same learning rates? Maybe different methods have different best learning rates.
4. The theoretical proof does not consider the data heterogeneity. Will the thresholds still converge under the data heterogeneity?

**Questions:**

1. See weakness 1, During local training with e < E, does the thresholds not be updated? Equation (5) and (6) only work for the e=E? And how the equation (6) is derived?
2. See weakness 2.
3. See weakness 3.
4. See weakness 4.
5. In experiments, E = 3 means local iterations = 3, or local epochs = 3? local iteration = 3 seems to be too small. Could you find other references to support this setting? Because many FL works set this as epochs [1].


[1] Communication-Efficient Learning of Deep Networks from Decentralized Data.
[2] SCAFFOLD: Stochastic Controlled Averaging for Federated Learning.
[3] Adaptive Federated Optimization.

---

> ### Author Response · Authors · 2023-11-16
>
> We appreciate the constructive comments. We provide our response to each comment below.
>
> >Q1 See weakness 1, During local training with e < E, does the thresholds not be updated? Equation (5) and (6) only work for the e=E? And how the equation (6) is derived?
>
> **A1**: Yes, when $e < E$, only model parameters are updated while thresholds are frozen. In the last epoch, i.e., $e = E$, we freeze model parameters and update thresholds using eq. (5) and (6). Hence, during $E-1$ epochs, model parameters can adapt to newly generated masks by recovering from pruning-induce noise. Then, in the last epoch, we give the thresholds performance feedback on the current sparse model. We believe that the explanation of the algorithm was not clear enough in the manuscript, and we will clarify it in the revised version.
>
> For the derivation of eq. (6), we define $S(\cdot)$ as a unit step function and $Q\_{k, ij}^{E-1, l} (t)) = |w\_{k, ij}^{E-1, l} (t)|  -  \tau_i^l(t)$. Then, we can derive eq. (6) as follows:
>
> $$ \frac{\partial F\_k (\tilde{w}\_k^{E-1}(t),  \tau(t) )}{\partial \tau_i^l(t)} = \sum\_{j=1}^{n} \frac{ \partial \tilde{w}\_{k, ij}^{E-1} (t) } { \partial \tau\_i^l (t) }   \frac{\partial F\_k (\tilde{w}\_k^{E-1} (t),  \tau(t)) }{\partial \tilde{w}\_{k, ij}^{E-1} (t) } =  \sum\_{j=1}^{n\_{in}^l}  \frac{ \partial \tilde{w}\_{k, ij}^{E-1} (t) } { \partial Q\_{k, ij}^{E-1, l} (t) } \frac{ \partial Q\_{k, ij}^{E-1, l} (t) } { \partial \tau\_i^l (t) } g\_k( \tilde{w}\_k^{E-1}(t))\_{ij}^l $$
> $$= \sum\_{j=1}^{n\_{in}^l} w\_{k, ij}^{E-1, l} (t) \frac{ \partial S(Q\_{k, ij}^{E-1, l} (t)) } { \partial Q\_{k, ij}^{E-1, l} (t) } (-1) g\_k( \tilde{w}\_k^{E-1}(t))\_{ij}^l  = -\sum\_{j=1}^{n\_{in}^l} w\_{k, ij}^{E-1, l} (t) g\_k( \tilde{w}\_k^{E-1}(t))\_{ij}^l ,$$
> where the last equation results from the fact that we used the identity straight-through estimator for $\frac{ \partial S(Q\_{k, ij}^{E-1, l} (t)) } { \partial Q\_{k, ij}^{E-1, l} (t) }$ and $w$ and $\tau$ are constrained to be within
> $[−1, 1]$ and $[0, 1]$ as described in Section. 3.2.1.
>
> >Q2. Comparing equation (10) and (11), authors concludes the relationship between the gradient direction and $\Delta \tau$
>  when  $w >0$ or $w<0$. However, Equation (9) is about the global $\tau$, while equation (10) is talking about local $\tau$. Could authors explain this in more details?
>
> **A2:** We would like to note that the thresholds in eq. (9) and (10) are both global thresholds. Specifically, in eq. (9) $\tau(t+1) = \sum\_{k \in S\_t} \tau\_k(t)$, $\tau(t+1)$ represents global thresholds and $\tau\_k(t)$ represents local thresholds updated by client $k$ at the last epoch. Eq. (10) is about the gradients of the loss with respect to the current global thresholds $\tau(t)$ as shown in eq. (5). Hence, the clients can obtain $\tau\_k(t)$ by updating $\tau(t)$ using eq. (10).
>
> The main purpose of Section 3.2.4 is to identify how to update model parameters for given $\Delta \tau(t) = \tau(t+1) - \tau(t)$. The update in Section 3.2.4 occurs after receiving the global thresholds $\tau(t+1)$, which is generated from eq. (9). If $\Delta \tau\_i^l < 0$, then the parameters connected to threshold $ \tau\_i^l $ are globally important. Otherwise, those parameters are less globally important. Hence, a client can deduce how to update its parameters for given $\Delta \tau(t)$. To this end, we used eq. (10) and (11) to explain the relationship between the sign of parameters, the sign of $\Delta \tau(t)$, and the gradient direction of the parameters.

---

> > ### Author Response · Authors · 2023-11-16
> >
> > >Q3. Experiment settings are not clear enough. The training dataset is split using Dirichlet sampling. For personalized FL, how are test datasets split and how the models are tested? Why all methods use the same learning rates? Maybe different methods have different best learning rates.
> >
> > **A3:** Test datasets are distributed using the same Dirichlet sampling used in distributing the training datasets. We calculated the accuracy by averaging each client’s model performance on its test datasets following [1, 2]. We averaged all the results over at least 10 different random seeds.  As stated by the reviewer, every method has different well working learning rates. But, we found that we didn't provide them in the manuscript. Please find the table for the learning rates of the baselines.
> >
> > |Algorithm| FMNIST | CIFAR-10 | CIFAR-100|
> > |:------:| :----: | :----:| :-----:|
> > |FedAvg| $lr = 0.001$ | $lr = 0.01$ | $lr = 0.1$|
> > |Sub-FedAvg|  $lr = 0.001$ | $lr = 0.001$ |$lr = 0.2$ |
> > |LG-FedAvg| $lr = 0.001$ | $lr = 0.005$ | $lr = 0.0015$|
> > |PruneFL| $lr = 0.01$ | $lr = 0.001$ | $lr = 0.05$|
> > |LotteryFL| $lr = 0.001$| $lr = 0.005$ | $lr = 0.01$|
> > |FedDST| $lr = 0.001$ | $lr = 0.01$ | $lr = 0.01$|
> > |FedPM| $lr = 0.15 $ |$lr = 0.1$ | $lr = 0.1$|
> >
> > We will add this table to the revised manuscript.
> >
> > [1] Li, Tian, et al. "Ditto: Fair and robust federated learning through personalization." International Conference on Machine Learning. PMLR, 2021.
> >
> > [2] Mugunthan, Vaikkunth, et al. "Fedltn: Federated learning for sparse and personalized lottery ticket networks." European Conference on Computer Vision. Cham: Springer Nature Switzerland, 2022.
> >
> > >Q4. The theoretical proof does not consider the data heterogeneity. Will the thresholds still converge under the data heterogeneity?
> >
> > **A4:**. We would like to note that we do consider the data heterogeneity in this paper, and we did not remove it from the theoretical analysis. However, the reason why the data heterogeneity does not appear explicitly in the proof is that we took a rather different approach compared to previous works [3] and [4]. In SpaFL, we do not have a global model and, thus, the clients only share thresholds and update them in the last epoch, i.e., e=E. The server generates global thresholds using the scheduled users’ gradients in the last epoch. Hence, in the proof, we showed that the sum of those gradients can be bounded based on the assumptions about the smoothness of the loss function and unbiasedness of the stochastic gradients of thresholds.  This approach eventually shows that global thresholds will converge. This result holds for data heterogeneity.
> >
> > [3] Karimireddy, Sai Praneeth, et al. "Scaffold: Stochastic controlled averaging for federated learning." International conference on machine learning. PMLR, 2020.
> >
> > [4] Reddi, Sashank J., et al. "Adaptive Federated Optimization." International Conference on Learning Representations. 2020.
> >
> > >Q5 In experiments, E = 3 means local iterations = 3, or local epochs = 3? local iteration = 3 seems to be too small. Could you find other references to support this setting? Because many FL works set this as epochs.
> >
> > **A5:**  $E=3$ means that scheduled clients perform 3 local epochs using their datasets. We found that we used ‘iteration’ in Section 3. We will clarify this in the revised version. For this choice of local epochs, we followed the settings as in [2] and [5]
> >
> > [5]: Isik, Berivan, et al. "Sparse Random Networks for Communication-Efficient Federated Learning." The Eleventh International Conference on Learning Representations. 2022.

---

> > > ### Comment · Reviewer_Kuri · 2023-11-20
> > > **Thanks for your responses**
> > >
> > > I enjoy reading the responses. Followed by previous questions and your responses, I have following questions:
> > > 1. How does Equation (8) lead to the conclusion "Therefore, if connected parameters were important, the corresponding threshold will decrease. Otherwise, the threshold will be increased to enforce sparsity." I didn't see the obvious connection between the Taylor expansion and the corresponding threshold.
> > > 2. the relationship between the gradient direction and  $\Delta\tau$ is not interpreted in responses. The $\tau$ can represent the parameter importance, and the gradient represents the updating direction of the weights. Would you explicitly elaborate more on how the gradient direction influences the parameter importance?
> > > 3. I disagree with that the data heterogeneity is implicitly considered. The bounded sum of gradients shows final convergence, but does not incorporate the influences of the data heterogeneity.

---

> > > > ### Author Response · Authors · 2023-11-21
> > > >
> > > > We appreciate the reviewer for the insightful questions and comments. We provide our response to each comment below.
> > > >
> > > > >Q1: How does Equation (8) lead to the conclusion "Therefore, if connected parameters were important, the corresponding threshold will decrease. Otherwise, the threshold will be increased to enforce sparsity." I didn't see the obvious connection between the Taylor expansion and the corresponding threshold.
> > > >
> > > > **A1:** In eq. (8), the importance of parameter $w\_{ij}^l$ is given as $F(w, \tau) - F(w, \tau; w\_{ij}^l=0) \approx g(w)\_{ij}^l w\_{ij}^l$. We can deduce that if $w$ was important, $F(w, \tau) - F(w, \tau; w=0)  \approx g(w)\_{ij}^l w\_{ij}^l < 0 $. Otherwise, $F(w, \tau) - F(w, \tau; w=0)  \approx g(w)\_{ij}^l w\_{ij}^l  > 0 $. Meanwhile, we update threshold $i$ in layer $l$ of client $k$ as $\tau\_{k,i}^l(t) = \tau\_i^l(t) - \eta(t) h\_{k, i}^l (\tilde{w}\_k^{E-1})  = \tau\_i^l(t) + \eta(t) \sum\_{j=1}^{n\_{in}^l} w\_{k, ij}^{E-1, l} (t)  g\_k( \tilde{w}\_k^{E-1}(t))\_{ij}^l $. For simplicity, we omitted the sparsity regularizer. We can see that if the connected parameters were important, then $\sum\_{j=1}^{n\_{in}^l} w\_{k, ij}^{E-1, l} (t)  g\_k( \tilde{w}\_k^{E-1}(t))\_{ij}^l $ will be negative, thereby decreasing the threshold. Otherwise, $\sum\_{j=1}^{n\_{in}^l} w\_{k, ij}^{E-1, l} (t)  g\_k( \tilde{w}\_k^{E-1}(t))\_{ij}^l $ will positive, and this will increase the threshold. We found that the two terms in eq. (8) were reversed in order in the manuscript. We will revise this to enhance the clarity.
> > > >
> > > > >Q2: the relationship between the gradient direction and $\Delta \tau$ is not interpreted in responses. The $\tau$ can represent the parameter importance, and the gradient represents the updating direction of the weights. Would you explicitly elaborate more on how the gradient direction influences the parameter importance?
> > > >
> > > > **A2:** From eq. (8), the importance of parameter $w\_{ij}^l$  is given by $F(w, \tau) - F(w, \tau; w\_{ij}^l=0) \approx g(w)\_{ij}^l w\_{ij}^l$. To elaborate how the gradient direction influences the parameter importance, first suppose that $w\_{ij}^l$ was important. Then, $F(w, \tau) - F(w, \tau; w\_{ij}^l=0) \approx g(w)\_{ij}^l w\_{ij}^l $ will be negative because $F(w, \tau; w\_{ij}^l=0)$ would be larger than $F(w, \tau) $.  Here, if $w\_{ij}^l > 0 $, then $g(w)\_{ij}^l$ should be negative. Otherwise, if $w\_{ij} < 0 $, then $g(w)\_{ij}^l$ should be positive. Hence, for given the sign of a parameter and its gradient direction, we can deduce the whether this parameter was important or not.
> > > >
> > > > Now, we explain the relationship between the gradient direction and $\Delta \tau$. Since $\Delta \tau(t+1)  = \frac{1}{K} \sum\_{k\in{S\_t}} g\_k(\tilde{w}\_k^{E-1}(t)) w\_k^{E-1}(t)$,  we can say that $\Delta \tau (t+1)$ is the aggregated parameter importance. Hence, once we know the sign of a parameter and the sign of $\Delta \tau (t+1)$, then we can deduce the corresponding sign of $g\_k$, which is the gradient direction of that parameter.

---

> > > > > ### Author Response · Authors · 2023-11-21
> > > > >
> > > > > >Q3: I disagree with that the data heterogeneity is implicitly considered. The bounded sum of gradients shows final convergence, but does not incorporate the influences of the data heterogeneity.
> > > > >
> > > > > **A3:** We agree with the reviewer that the data heterogeneity should be considered in the final convergence. Although we did not remove the data heterogeneity in this paper, we provide the revised version of our analysis to show the data heterogeneity explicitly.
> > > > >
> > > > > Since the loss function $F\_k (\cdot)$ is $M$-smooth for $\tau$, we have the following [1] $F\_k(\tilde{w}\_k^{E-1}(t), \tau(t)) - F\_k(\tilde{w}\_k^{E-1}(t), \tau\_k(t)) \leq  \langle \nabla\_{\tau} F\_k(\tilde{w}\_k^{E-1}(t), \tau-\tau\_k(t) \rangle + \frac{1}{2M} ||  \nabla\_{\tau} F\_k(\tilde{w}\_k^{E-1}(t), \tau(t)) -  \nabla\_{\tau\_k} F\_k(\tilde{w}\_k^{E-1}(t), \tau\_k(t))||^2 $
> > > > > $\leq \frac{\gamma(t)}{2} ||\nabla\_{\tau} F\_k(\tilde{w}\_k^{E-1}(t), \tau(t)) ||^2 + \frac{1}{2\gamma(t)} ||\tau(t) - \tau\_k(t)||^2 +  \frac{1}{2M} ||  \nabla\_{\tau} F\_k(\tilde{w}\_k^{E-1}(t), \tau(t)) -  \nabla\_{\tau\_k} F\_k(\tilde{w}\_k^{E-1}(t), \tau\_k(t))||^2$.
> > > > >
> > > > > We bound  $F\_k(\tilde{w}\_k^{E-1}(t), \tau(t)) - F\_k(\tilde{w}\_k^{E-1}(t), \tau\_k(t))$ in eq. (28) using the above inequality. Then by following eq. (29) and (30), we have the following revised convergence.
> > > > >
> > > > > $\frac{1}{NT} \sum\_{t=0}^{T-1}  \mathbb{E} ||\sum\_{k=1}^{N} \nabla\_{\tau} F\_k (\tilde{w}\_k^{E-1}(t), \tau(t))||^2
> > > > > 	\leq 	\sum\_{t=0}^{T-1} \sum\_{k=1}^{N} \frac{1}{MNT}  \underbrace{\mathbb{E}||  \nabla\_{\tau} F\_k(\tilde{w}\_k^{E-1}(t), \tau(t)) -  \nabla\_{\tau\_k} F\_k(\tilde{w}\_k^{E-1}(t), \tau\_k(t))||^2}\_{(A)} $
> > > > > $  + \sum\_{t=0}^{T-1} \frac{2\alpha \eta(t)}{T \gamma(t)} \bigg(
> > > > > 	1 - M \eta(t)(1-\alpha)  \bigg) ||\exp(-\tau(t))||^2  + \sum\_{t=0}^{T-1}  \sum\_{k=1}^{N} \frac{1}{\gamma(t) NT}||\tau(t) - \tau\_k(t)||^2  $
> > > > > $+  \sum\_{t=0}^{T-1}  \sum\_{k=1}^{N} \frac{2M^2\eta(t)^2 \eta\_{in}^{max}}{\gamma(t)NT} \mathbb{E} F\_k(\tilde{w}\_k^{E-1}(t), \tau(t)). $
> > > > >
> > > > > Here, $(A)$ captures the difference between the gradient of global thresholds and that of local thresholds, which are updated using clients' private dataset. As the data heterogeneity increases, the gradients of local thresholds will deviate from that of global thresholds further, thereby damaging the convergence. We believe that the term $(A)$ can capture the data heterogeneity of clients.
> > > > >
> > > > >
> > > > >
> > > > > [1] Zhou, Xingyu. "On the fenchel duality between strong convexity and lipschitz continuous gradient." arXiv preprint arXiv:1803.06573 (2018).

---

> > > > > > ### Comment · Reviewer_Kuri · 2023-11-22
> > > > > > **Thanks for your further responses**
> > > > > >
> > > > > > Thanks for your clarification about the questions. I'd like to raise my score as 6 based on the insights and new convergence analysis.

---

> > > > > > > ### Author Response · Authors · 2023-11-22
> > > > > > >
> > > > > > > We appreciate the reviewer for raising the score.

---

### Official Review · Reviewer_R775 · 2023-10-31

**Soundness:** 3 good
**Presentation:** 3 good
**Contribution:** 2 fair
**Rating:** 5
**Confidence:** 3

**Summary:**

This paper introduces a method to mitigate the communication and computation overhead in FL. It employs a threshold-based approach to simultaneously optimize sparse masks and model parameters, resulting in reduced communication costs and the attainment of better personalized models. The paper offers a theoretical analysis regarding the convergence of the proposed method, while empirical results further confirm its efficacy.

**Strengths:**

1. The proposed approach employs a straightforward method based on transmitting threshold to effectively reduce communication bandwidth while surpassing the accuracy of personalized models over baseline methods.

2. The paper substantiates the effectiveness of the proposed method through comprehensive empirical evaluations and theoretical analysis.

3. The paper is well-written and easy to read.

**Weaknesses:**

1. Some implementation details need further clarification. (Q1)

2. The intuition from the theory needs more elaboration. (Q2)

3. This study primarily concentrates on personalized federated learning, where no global model is trained. It would enhance clarity if the authors could differentiate between personalized federated learning (pFL) and federated learning (FL), as the paper references FL multiple times, which typically involves a global model.

**Questions:**

1. During the local training for parameters and thresholds, if the gradients are calculated for every weight and applied with the binary mask, then how does it help save the computation overhead? Or if the gradients are calculated w.r.t. the sparse weights, how is it achieved in practice?

2. The interpretation of the third term in Theorem 1 is not straightforward. The loss function $F_k$ is not bounded in the paper, and it can potentially assume arbitrarily large values, rendering the third term of Theorem 1 indeterminate. How to understand Theorem 1 as a valid convergence bound?

---

> ### Author Response · Authors · 2023-11-16
>
> We appreciate the reviewer for the constructive feedback. We provide our response to each comment below.
>
> >Q1. During the local training for parameters and thresholds, if the gradients are calculated for every weight and applied with the binary mask, then how does it help save the computation overhead? Or if the gradients are calculated w.r.t. the sparse weights, how is it achieved in practice?
>
> **A1:** During the backpropagation, for each layer, we calculate the gradients for the input activations and the gradients for the weights. In SpaFL, we are able to reduce the number of FLOPs to calculate the gradients for input activations by convolving sparse weight tensors with the gradients of the output activations. In particular, we sparsified the weight tensors before performing convolutions by detaching them from the computation graph. Below, we provide a detailed explanation about how we reduce the computation overhead of the backpropagation.
>
> Consider a convolutional layer with an input tensor $𝑋 ∈ R^{𝑁×𝐶×𝑋×𝑌},$ parameter tensor $𝑊 ∈ R^{𝐹 ×𝐶×𝑅×𝑆}$, and output tensor $𝑂 ∈ R^{𝑁×𝐹 ×𝐻×𝑊}$. Here, the input tensor $X$ consists of $N$ number of samples, each of which has $X×Y$ dimension. The parameter tensor $W$ has $F$ filters of $C$ channels with kernel size $R × S$. Assume that $𝑊$ is pruned, and $𝜌$% of the parameters are active. The output tensor $O$ will have $F$ output channels with dimension $H × W$ for $N$ samples. We can calculate the number of FLOPs for the gradients of input activations by convolving the gradient of output activations with the parameter matrix $W$. Hence, it can be approximated to $𝜌 × 𝑁 × (𝐹 × 𝑅 × 𝑆) × 𝐶 × 𝑋 × 𝑌$. Since we used a sparse tensor $𝑊$ during the forward propagation, we can reduce the number of FLOPs for calculating the gradients of input activations by $𝜌$. Then, we can calculate the gradients of the weights by convolving the input activation with the output gradient. This can be approximately given by $(N × X × Y ) × F × C× R × S$. Since we did not sparsify the activations, there is no reduction of FLOPs in this calculation. Therefore, the total number of FLOPs for the backpropagation in SpaFL can be given by $(1 + 𝜌) × 𝑁 × 𝐹 × 𝑅 × 𝑆 × 𝐶 × 𝑋 × 𝑌$. Meanwhile, without sparsity, the number of FLOPs for the backpropagation will be $2 × 𝑁 × 𝐹 × 𝑅 × 𝑆 × 𝐶 × 𝑋 × 𝑌$. Hence, we can reduce the computing overhead by $\frac{1+𝜌}{2}$ in the backpropagation compared to a baseline that does not use sparse weights. As shown in Fig. 2 in the manuscript, model densities in SpaFL usually go below $10$% after 200 rounds. Hence, we can reduce the computation overhead in the backpropagation around $30$% compared to dense baselines during the training.

---

> > ### Author Response · Authors · 2023-11-16
> >
> > >Q2. The interpretation of the third term in Theorem 1 is not straightforward. The loss function $F\_k$ is not bounded in the paper, and it can potentially assume arbitrarily large values, rendering the third term of Theorem 1 indeterminate. How to understand Theorem 1 as a valid convergence bound?
> >
> > **A2:** In Section 3.2.2, we constrained the model parameters w and thresholds $\tau$ to be within $[-1,1]$ and $[0,1]$, respectively. Therefore, if input-output pairs ${x,y}$ are drawn from some bounded domain, the loss function $F\_k$ can be obviously bounded by a finite real value. Assuming a bounded loss has been also widely adopted in key papers such as [1] and [2].
> >
> > However, we also agree with the reviewer that the loss function can still be an arbitrary large value. To further address this comment, we next provide a revised analysis that can help show that our Theorem 1 can be presented without using $F\_k$ in the third term. Before this analysis, we need to make the following assumption.
> >
> > (Bounded stochastic gradient) There exists $G \geq 0$ for client $k, \forall k$ such that $\mathbb{E} || g\_k(w\_k)||^2 \leq G^2$.
> > We would like to note that this assumption is widely adopted in previous works such as [2, 3, 4].
> >
> > Then, we can revise the term $B$ in eq. (25) in the proof as follows
> > $$ B = \frac{M \eta(t)^2}{2} \mathbb{E}|| h\_k(\tilde{w}\_k^{E-1}(t)) ||^2 =  \frac{M \eta(t)^2}{2} \mathbb{E}  \sum\_{l=1}^{L}\sum\_{i=1}^{n\_{out}^l} || \sum\_{j=1}^{n\_{in}^l} { g\_k(\tilde{w}\_k^{E-1} (t)) }\_{ij}^{l} w\_{k, ij}^{E-1, l}(t) ||^2 $$
> > $$\leq \frac{M \eta(t)^2}{2} \mathbb{E}  \sum\_{l=1}^{L} \sum\_{i=1}^{n\_{out}^l} n\_{in}^l \sum\_{j=1}^{n\_{in}^l} || { g\_k(\tilde{w}\_k^{E-1}(t)) }\_{ij}^{l} w\_{k, ij}^{E-1, l}(t) ||^2 $$
> > $$\leq \frac{M \eta(t)^2 n\_{in}^{max} }{2} \mathbb{E}  \sum\_{l=1}^{L} \sum\_{i=1}^{n\_{out}^l} \sum\_{j=1}^{n\_{in}^l} || { g\_k(\tilde{w}\_k^{E-1}(t)) }\_{ij}^{l} w\_{k, ij}^{E-1, l}(t) ||^2 $$
> > $$\overset{(a)}{\leq}    \frac{M \eta(t)^2 n\_{in}^{max} }{2} \mathbb{E}  \sum\_{l=1}^{L} \sum\_{i=1}^{n\_{out}^l} \sum\_{j=1}^{n\_{in}^l} || { g\_k(\tilde{w}\_k^{E-1}(t)) }\_{ij}^{l} ||^2 $$
> > $$ =  \frac{M \eta(t)^2 n\_{in}^{max} }{2} \mathbb{E} || g\_k(\tilde{w}\_k^{E-1}(t)) ||^2 \leq  \frac{M \eta(t)^2 n\_{in}^{max} }{2} G^2,$$
> > where $n\_{in}^{max}$ is the largest number of parameters connected a filter/neuron in a given model, (a) is from $|w| \leq 1$ and the last inequality results from the newly introduced assumption. Based on the new bound of $B$, we can derive the following inequality:
> > $$\mathbb{E} F\_k(\tilde{w}\_k^{E-1}(t), \tau\_k(t)) \leq \mathbb{E} F\_k (\tilde{w}\_k^{E-1}(t), \tau(t)) - \eta(t)
> > \bigg(1 - \frac{\alpha ( 1- M \eta(t))} {2} \bigg) ||\nabla_\{\tau} F\_k(\tilde{w}\_k^{E-1}(t), \tau(t))||^2$$
> > $$+\frac{\alpha \eta(t) (1 - M \eta(t)(1-\alpha )}{2} ||\exp(-\tau(t))||^2 + \frac{M \eta(t)^2 n\_{in}^{max}}{2} G^2.$$
> > By following eq. (28) and (29), we have the following convergence bound
> > $$\frac{1}{NT} \sum\_{t=0}^{T-1}  \mathbb{E} ||\sum\_{k=1}^{N} \nabla\_{\tau} F\_k (\tilde{w}\_k^{E-1}(t), \tau(t))||^2
> > 	\leq 	\sum\_{t=0}^{T-1} \sum\_{k=1}^{N}  \frac{\mathbb{E} |F\_k (\tilde{w}\_k^{E-1}(t), \tau(t)) -  F\_k (\tilde{w}\_k^{E-1}(t), \tau\_k(t))|}{N T \gamma(t)}$$
> > $$  + \sum\_{t=0}^{T-1} \frac{\alpha \eta(t)}{T \gamma(t)} \bigg(
> > 	1 - M \eta(t)(1-\alpha)  \bigg) ||\exp(-\tau(t))||^2  + \sum\_{t=0}^{T-1} \frac{M \eta(t)^2 n\_{in}^{max} } {2 T \gamma(t)} G^2.$$
> >
> > Now, since $G$ is a constant and the learning rate $\eta(t)$ can be a decreasing function. Then, we can show that the third term is also a decreasing function as follows
> >
> > $$\frac{M \eta(t)^2 n\_{in}^{max} } {2T \gamma(t)} G^2 = \frac{M \eta(t)^2 n\_{in}^{max} } {T(2\eta(t) + \alpha M \eta(t)^2 - \alpha \eta(t))  } G^2 = \frac{M n\_{in}^{max} } { \frac{T(2-\alpha) }{\eta(t)} + T\alpha M } G^2.$$
> > Since we assumed $0\leq \alpha \leq 1$, the third term is decreasing as $\eta(t)$ decreases. Therefore, we believe that we improved the interpretation of the third term in this revised analysis.
> >
> > [1] Deng, Yuyang, Mohammad Mahdi Kamani, and Mehrdad Mahdavi. "Adaptive personalized federated learning." arXiv preprint arXiv:2003.13461 (2020).
> >
> > [2] Mohri, Mehryar, Gary Sivek, and Ananda Theertha Suresh. "Agnostic federated learning." International Conference on Machine Learning. PMLR, 2019.
> >
> > [3] Li, Xiang, et al. "On the Convergence of FedAvg on Non-IID Data." International Conference on Learning Representations. 2019.
> >
> > [4] Deng, Yuyang, Mohammad Mahdi Kamani, and Mehrdad Mahdavi. "Distributionally robust federated averaging." Advances in neural information processing systems 33 (2020): 15111-15122.

---

> > > ### Author Response · Authors · 2023-11-16
> > >
> > > >Q3. This study primarily concentrates on personalized federated learning, where no global model is trained. It would enhance clarity if the authors could differentiate between personalized federated learning (pFL) and federated learning (FL), as the paper references FL multiple times, which typically involves a global model.
> > >
> > > **A3:** We agree with the reviewer that we can improve the clarity of the manuscript by explicitly differentiating personalized federated learning (pFL) and federated learning (FL). We will revise the manuscript by mentioning that SpaFL considers a pFL scenario. This is a straightforward revision.

---

### Official Review · Reviewer_HLPa · 2023-10-31

**Soundness:** 2 fair
**Presentation:** 3 good
**Contribution:** 3 good
**Rating:** 6
**Confidence:** 4

**Summary:**

This paper proposes SpaFL to tackle the communication cost problem in federated learning. To find the sparse mask for the model, authors introduce a new parameter called threshold ($\tau$). This parameter indicates if a weight in the model is active or nonactive, hence can reduce the density of the model. In each round, the clients find the current mask based on $\tau$, then update their local weight, and finally update the local $\tau$. After the local step, the server aggregates the client's $\tau$ and transmits the new value to all the clients. Then, all the clients update their weights accordingly.

**Strengths:**

* The problem is well-motivated.
* Authors provides theoretical proof for the convergence of their method.
* The method is novel and saves uplink communication costs for the clients.
* The solution is novel.

**Weaknesses:**

* What happens if clients do not receive the server update due to unavailability? It is specifically important as the solutions is designed  for resource-constrained cross-device FL, where clients are only sometimes available.
* How do non-participant clients update their model?
* Is there any global model available?
* The author should include a comparison with prior works that adapt sparse learning in FL, such as [8,9,22,24,25,26] (references are from this paper). Some of these methods can reach high sparsities comparable to the 1% communication cost of SpaFL.
* How does the server or clients control the density of the models.
* How does SpaFL perform when the global model is denser (for example ResNet18)?

**Questions:**

* The questions can be found in weaknesses.

---

> ### Author Response · Authors · 2023-11-18
>
> We appreciate the reviewer for the constructive comments. We provide our response to each comment below.
>
> >Q1: What happens if clients do not receive the server update due to unavailability? It is specifically important as the solutions is designed for resource-constrained cross-device FL, where clients are only sometimes available.
>
> **A1:** We agree with the reviewer that clients may not always be able to receive global thresholds due to their constrained resources. To demonstrate that SpaFL is also compatible with intermittent availability, we did comparison between SpaFL and a baseline where only a small subset of clients can receive the server update. This baseline can capture the unavailability of some clients who cannot participate in the training at a given time. Below, we provide an empirical comparison between SpaFL and the baseline with the scheduling size of 10.
>
> | Algorithms | FMNIST | CIFAR-10 | CIFAR-100|
> |:------:|:--------:|:-------:|:--------:|
> |SpaFL|$90.31\pm0.35$ | $73.85\pm2.80$| $38.80\pm1.10$|
> |SpaFL + unavailability|$90.24\pm0.08$| $73.45\pm3.19$| $38.65 \pm 0.43$|
>
> For SpaFL, the communications costs are $1.0208$ Gbit, $2.4956$ Gbit, and  $0.7674$ Gbit, for FMNIST, CIFAR-10, and CIFAR-100, respectively. For the baseline, the communications costs are $0.1856$ Gbit, $0.4537$ Gbit, and  $0.1395$ Gbit, for FMNIST, CIFAR-10, and CIFAR-100, respectively. We can see that the performance loss of the baseline is very small. Moreover, the baseline used only $20$% of the communication costs used by SpaFL. Therefore, this result demonstrates the compatibility of SpaFL with a scenario where only a subset of clients is available and receives the server update.
>
> >Q2. How do non-participant clients update their model?
>
> **A2:** As demonstrated by response **A1**, intermittently non-participating clients can update their model when they become available and receive server update. If a client is not participating in the training at a given time, then we can proceed with the SpaFL algorithm with a subset of participating clients. However, if some clients do not participate in the training permanently, then they cannot update their model in SpaFL. However, we believe this is the case for most of FL algorithms and not unique to SpaFL.
>
> >Q3: Is there any global model available?
>
> **A3:** Since SpaFL personalizes clients' models, it does not have a global model. Instead, clients share global thresholds. Since global thresholds are aggregated parameter importance, clients can know which neuron/filter is globally important, thereby learning how to prune their model.
>
> However, SpaFL can also be extended to share some model parameters between clients. Clients can transmit their unpruned parameters to the server as done in [8,9,22,24,25,26] (references are from the manuscript) along with their thresholds. Then, the server can generate a global model by aggregating the received parameters and thresholds. Hence, we extend SpaFL to enable clients to share both global model and thresholds.

---

> > ### Author Response · Authors · 2023-11-18
> >
> > >Q4: The author should include a comparison with prior works that adapt sparse learning in FL, such as [8,9,22,24,25,26] (references are from this paper). Some of these methods can reach high sparsities comparable to the 1% communication cost of SpaFL.
> >
> > **A4:** We agree with the reviewer that a comparison between SpaFL and works with sparse learning will be important to demonstrate the low communication and computing costs of SpaFL. Hence, we compare SpaFL with two sparse learning baselines [1, 2] as the reviewer suggested. The work in [1] adopted sparse learning in a personalized FL setting, and [2] trained a global sparse model across clients. Here, we would like to note that we already showed the performance of [2] in our manuscript. For [1], we fixed the density of models to 50% following their methodology. We used the same network architectures used in our manuscript for this experiment and set learning rates of [1] as 0.001, 0.01, and 0.1 for FMNIST, CIFAR-10, CIFAR-100, respectively. For CIFAR-100, we decayed the learning rate by 0.998 per each round. For other hyperparameters (e.g., batch size, local epoch $E$, scheduling size $K$), we used the same settings in our manuscript for each dataset.  Below, we provide an empirical comparison.
> >
> > | Algorithms | FMNIST | CIFAR-10 | CIFAR-100|
> > |:------:|:--------:|:-------:|:--------:|
> > |SpaFL|$\mathbf{90.31\pm0.35}$  | $\mathbf{73.85\pm2.80}$| $\mathbf{38.80\pm1.10}$|
> > |FedSpa [1]|$89.30\pm0.20$| $67.63\pm0.05$| $36.52 \pm 0.03$|
> > |FedDST [2] | $84.46\pm0.14$ | $60.18\pm0.03$ |$22.26\pm 0.14$|
> >
> > We also provide the amount of used computing and communications below. 'FLOP' is the number of used FLOPs and 'Comm' is the number of communicated bits between clients and a server during the training.
> >
> > | Algorithms |       FMNIST (FLOP, Comm)         | CIFAR-10 (FLOP, Comm) | CIFAR-100 (FLOP, Comm)|
> > |:------:|:--------:|:-------:|:--------:|
> > |SpaFL|  $\mathbf{(2.72 \times10^{11}, 1.02 Gbit )} $| $\mathbf{(1.59\times10^{12}, 2.50 Gbit)}$ | $\mathbf{(2.84 \times10^{11}, 0.77 Gbit)}$ |
> > |FedSpa [1]|$(5.25\times10^{11}, 55.2 \text{Gbit})$| $(4.29 \times 10^{12}, 129  \text{Gbit}$| $(9.27 \times 10^{11},  10.2  \text{Gbit} )$|
> > |FedDST [2] | $(2.89\times10^{11}, 74.5 \text{Gbit})$ | $(3.44\times10^{12}, 139  \text{Gbit})$ |$(8.84\times10^{11}, 13.3  \text{Gbit})$|
> >
> > Hence, we can see that SpaFL achieves better performance with less resources compared to both sparse learning baselines.
> >
> > >Q5: How does the server or clients control the density of the models.
> >
> > **A5:** In SpaFL, we control the density of models through the regularizer $\alpha$. As $\alpha$ increases, we enforce more sparsity into models. Unlike works in [8,9,22,24,25,26], which fix the density of models, SpaFL makes models as sparse as possible while balancing the performance and sparsity. This is a meaningful advantage because it enables clients to learn optimal sparse structures that can minimize the performance loss. We can see the improvements in performance and costs achieved by SpaFL in comparsion to the approaches in [8,9,22,24,25,26] from the response **A4**.
> >
> > >Q6: How does SpaFL perform when the global model is denser (for example ResNet18)?
> >
> > **A6:**
> >
> > We trained ResNet-18 [3] on CIFAR-100 distributed over 100 clients in a non-iid fashion following Dirichlet (0.1) distribution for $1000$ rounds. The initial learning rate was 0.01 and decayed to 0.0001 by multiplying 0.993 at each round. We set sparsity regularizer $\alpha = 0.0005$, local epoch $E=2$, scheduling size $K=10$, and batch size of 64. Below, we provide an empirical result.
> >
> > | Algorithms | Acc | FLOP (e+13) | Comm (Gbit)|
> > |:------:|:--------:|:-------:|:--------:|
> > |SpaFL|$\mathbf{44.35\pm0.39}$  | $\mathbf{4.5288}$| $\mathbf{8.448}$|
> > |FedAvg|$36.50\pm0.42$| $ 16.626$| $3570.9$|
> >
> > From the above table, we can see that SpaFL still works well using significantly less computing and communication resources compared to the dense baseline (FedAvg).
> >
> >
> > [1] Liu, Xiaofeng, et al. "Sparse Personalized Federated Learning." IEEE Transactions on Neural Networks and Learning Systems (2023).
> >
> > [2] Bibikar, Sameer, et al. "Federated dynamic sparse training: Computing less, communicating less, yet learning better." Proceedings of the AAAI Conference on Artificial Intelligence. Vol. 36. No. 6. 2022.
> >
> > [3] He, Kaiming, et al. "Deep residual learning for image recognition." Proceedings of the IEEE conference on computer vision and pattern recognition. 2016.

---

> > > ### Comment · Reviewer_HLPa · 2023-11-22
> > > **Respond to rebuttal**
> > >
> > > I want to thank the author for their comprehensive response; I increased my score to 6.
> > >
> > > It would be helpful if the authors could explain the client-side update algorithm for clients who were unavailable for several rounds, especially since the clients are maintaining their local state and there is no global model.

---

> > > > ### Author Response · Authors · 2023-11-22
> > > >
> > > > We appreciate the reviewer for the constructive feedback and raising the score. We provide our response to the comment below.
> > > >
> > > > >Q1: It would be helpful if the authors could explain the client-side update algorithm for clients who were unavailable for several rounds, especially since the clients are maintaining their local state and there is no global model.
> > > >
> > > > **A1:** We explain the client-side update algorithm for clients who were unavailable for several rounds. As discussed in the previous response, unavailable clients cannot receive the server update and cannot participate in the training. Once they become available, they receive the current global thresholds from the server. Consequently, they update their models using eq. (12) in the manuscript by extracting the aggregated parameter importance from the current global thresholds. Then, they perform local training for $E$ epochs.

---

### Official Review · Reviewer_Ggae · 2023-11-07

**Soundness:** 3 good
**Presentation:** 3 good
**Contribution:** 3 good
**Rating:** 6
**Confidence:** 3

**Summary:**

The authors propose a novel federated learning approach with sparse personalized client models. The main technical contribution is the reduction of communication overhead by only communicating the thresholds used to determine non-zero parameters and the reduction of computation cost by joint optimization of thresholds and sparse client models. The convergence of the approach is theoretically analyzed and experiments on a range of datasets demonstrate improvements over several prior works.

**Strengths:**

1. The proposed approach will significantly reduce communication cost since it only involves communicating one threshold per neuron and number of neurons is much less than the number of parameters.

2. The empirical results also show a significant reduction in FLOPs due to the sparsity of the models being optimized on the clients.

**Weaknesses:**

1. Some aspects of the algorithm are not clearly explained. It is not clear why the regularizer in (4) is chosen over other options. It is also not clear why the second update of the client models in 3.2.4 is necessary because technically it should be possible to continue training the model with the new global thresholds as described in 3.2.2.

2.The sparsification is unstructured and thus may not actually lead to reduction in computation cost or latency due to inefficient utilization of the hardware. Since there are already several works on sparse FL I believe it is important to now start considering hardware performance etc to truly differentiate from prior work.

**Questions:**

1. In addition to an intuitive justification for the regularizer in (4) and the second update in 3.2.4 can you also provide an empirical comparison with alternate regularizers?

2. Likewise can you also provide a comparison with a baseline which does not use the update in 3.2.4 but instead just directly continues training the model with the new global thresholds?

3. Can you provide a derivation for (6) and (11)?

4. From (10) and (11) if the gradient direction of $w$ is opposite to that of its connected threshold if $w>0$ then shouldn't the gradient direction of $w$ and $\Delta \tau$ be the opposite if $w>0$ and not same as is claimed in the paragraph after 11? Please clarify.

5. What is the model density of the baselines in Table 1? I do not see it presented in the table.

6. Do you have any thoughts on how the sparsification approach described herein could be made structured and thus more hardware efficient?

---

> ### Author Response · Authors · 2023-11-15
>
> Thank you for the constructive comments. We provide our response to each comment below.
>
> >Q1: In addition to an intuitive justification for the regularizer in (4) and the second update in 3.2.4 can you also provide an empirical comparison with alternate regularizers?
>
>  **A1**:  The main reason for choosing the regularizer  $R=\exp(-||\tau||)$ in (4) is that the exponential function goes to zero asymptotically as thresholds increase. This is reasonable because it can penalize low thresholds without making them become extremely large. We can also use a similar regularizer that can give penalty for low thresholds without encouraging them to become a too large value. For example, a regularizer $R' = \frac{1}{||\tau|| + 0.1}$ satisfies such property. Below, we provide an empirical comparison between SpaFL and the baseline that uses $R'$ as below.
> |Algorithm| FMNIST| CIFAR-10| CIFAR-100|
> |:-------------------:|:-------------------:|:------:|:------:|
> |SpaFL|    $\boldsymbol{90.31\pm0.34}$    |  $\boldsymbol{73.85 \pm 2.80}$ | $\boldsymbol{38.80 \pm 1.10}$|
> |Baseline| $89.92 \pm 0.25$| $73.62 \pm 3.42$ | $38.73 \pm 0.53$ |
>
> For the regularizer $R$, the achieved model densities are $5.36$%, $7.57$% and $7.38$% for FMNIST, CIFAR-10, and CIFAR-100, respectively. Meanwhile, with $R'$, the model densities are $4.85$%, $5.60$%, and $8.01$% for FMNIST, CIFAR-10, and CIFAR-100, respectively. We can see that both regularizers work well with SpaFL. Hence, SpaFL is compatible with any regularizers that can penalize the magnitude of thresholds asymptotically.
>
> >Q2: Likewise can you also provide a comparison with a baseline which does not use the update in 3.2.4 but instead just directly continues training the model with the new global thresholds?
>
> **A2**: The main purpose of Section 3.2.4 is to update the model parameters by extracting the aggregated importance from global thresholds. By investigating the difference between two consecutive global thresholds and parameters, clients can deduce which parameters are globally important.  Although clients perform their training using the received global thresholds as described in Section 3.2.2, this additional update provides meaningful performance gains. We provide an empirical comparison between SpaFL and the baseline that does not use the update in 3.2.4 as below
> |Algorithm| FMNIST| CIFAR-10| CIFAR-100|
> |:-------------------:|:-------------------:|:------:|:------:|
> |SpaFL|    $\boldsymbol{90.31\pm0.34}$    |  $\boldsymbol{73.85 \pm 2.80}$ | $\boldsymbol{38.80 \pm 1.10}$|
> |Baseline| $90.02 \pm 0.30$| $72.92 \pm 2.15$ | $38.06 \pm 0.80$ |
>
> We can see that the update in Section 3.2.4. can provide a clear improvement compared to the baseline by utilizing the aggregated parameter importance from global thresholds.
>
> >Q3: Can you provide a derivation for (6) and (11)?
>
> **A3**:  To derive (6), we start from the gradient of the loss with respect to a threshold, as follows:
>
> $$ \frac{\partial F\_k (\tilde{w}\_k^{E-1}(t),  \tau(t) )}{\partial \tau_i^l(t)} = \sum\_{j=1}^{n} \frac{ \partial \tilde{w}\_{k, ij}^{E-1} (t) } { \partial \tau\_i^l (t) }   \frac{\partial F\_k (\tilde{w}\_k^{E-1} (t),  \tau(t)) }{\partial \tilde{w}\_{k, ij}^{E-1} (t) }=  \sum\_{j=1}^{n\_{in}^l}  \frac{ \partial \tilde{w}\_{k, ij}^{E-1} (t) } { \partial Q\_{k, ij}^{E-1, l} (t) } \frac{ \partial Q\_{k, ij}^{E-1, l} (t) } { \partial \tau\_i^l (t) } g\_k( \tilde{w}\_k^{E-1}(t))\_{ij}^l $$
> $$= \sum\_{j=1}^{n\_{in}^l} w\_{k, ij}^{E-1, l} (t) \frac{ \partial S(Q\_{k, ij}^{E-1, l} (t)) } { \partial Q\_{k, ij}^{E-1, l} (t) } (-1) g\_k( \tilde{w}\_k^{E-1}(t))\_{ij}^l = -\sum\_{j=1}^{n\_{in}^l} w\_{k, ij}^{E-1, l} (t) g\_k( \tilde{w}\_k^{E-1}(t))\_{ij}^l ,$$
> where $Q\_{k, ij}^{E-1, l} (t)) = |w\_{k, ij}^{E-1, l} (t)|  -  \tau_i^l(t)$, $S(\cdot)$ is a unit step function, $\tilde{w} = w \cdot S(Q(w))$. The last equation results from the fact that we used the identity straight-through estimator and $w$ and $\tau$ is constrained to be within
> $[−1, 1]$ and $[0, 1]$ as described in Section 3.2.1 and 3.2.2.
>
> The derivation of eq. (11) is similar to the above. We start from the gradient of the loss with respect to a parameter $w$ as follows
> $$ \frac{\partial F\_k (\tilde{w}\_k(t), \tau(t) )} {\partial w\_k(t)}  =  \frac{\partial \tilde{w}\_k(t)}{\partial w\_k(t)}  \frac{  F\_k (\tilde{w}\_k(t), \tau(t) ) } {\partial \tilde{w}\_k(t) }= \frac{\partial \tilde{w}\_k(t) }{ \partial Q\_k(t) } \frac{\partial Q\_k(t)}{\partial w\_k(t)} g\_k( \tilde{w}\_k(t) )$$
> $$ = w\_k(t) \frac{\partial S(Q\_k(t))}{\partial Q\_k(t)} \frac{\partial |w\_k(t)|}{\partial w\_k(t)}  g\_k( \tilde{w}\_k(t) ) = |w\_k(t)| g\_k( \tilde{w}\_k(t) ).$$
> This completes the derivation of eq. (11).

---

> > ### Author Response · Authors · 2023-11-15
> >
> > > Q4: From (10) and (11) if the gradient direction of $w$ is opposite to that of its connected threshold if $w>0$ then shouldn't the gradient direction of $w$ and $\Delta \tau$ be the opposite if $w>0$ and not same as is claimed in the paragraph after 11? Please clarify.
> >
> > **A4**: Our claim is that if $w>0$, the sign of $\Delta \tau$ and the gradient direction of $w$ will be the same. This is because $\Delta \tau(t) = \tau(t+1) - \tau(t) = -h(\tilde{w}(t)) = g(w(t))w(t)$. Meanwhile, the gradient of $w$ is $g(w(t))|w(t)|$. Therefore, if $w>0$, the sign of $\Delta \tau$ and the gradient direction of $w$ will be the same. Otherwise, they will be opposite. We agree with you that the explanation for this aspect in Section 3.2.4. in the manuscript can be confusing, and we will clarify it in the revised version.
> >
> > >Q5: What is the model density of the baselines in Table 1? I do not see it presented in the table.
> >
> >
> > **A5**: As described in Section 5.2, the target model density of the baselines that use pruning is set to 0.5 following the configurations therein.
> >
> > >Q6: Do you have any thoughts on how the sparsification approach described herein could be made structured and thus more hardware efficient?
> >
> > **A6**: Yes, SpaFL can be readily extended to perform structured pruning. Specifically, we can prune entire parameters connected to a filter/neuron.  In SpaFL, we defined a threshold for each filter/neuron and pruned each connected parameter if its magnitude is smaller than that of the threshold. This approach leads to unstructured sparsity.  We can endow SpaFL with structured sparsity by calculating the average magnitude of all parameters connected to their filter/neuron and pruning all of them if their average magnitude is smaller than their threshold. Hence, SpaFL can be extended to do filter/neuron-wise pruning. In fact, we have implemented SpaFL with structured sparsity following your comments, and you can see the performance in the following table.
> >
> > |Algorithm| FMNIST| CIFAR-10| CIFAR-100|
> > |:-------:|:--------:|:------:|:------:|
> > |Structured SpaFL|    $\boldsymbol{89.71\pm0.44}$    |  $\boldsymbol{70.05 \pm 2.65}$ | $\boldsymbol{33.68 \pm 3.48}$|
> > |Sub-FedAvg| $89.29\pm0.69$ | $70.05 \pm 1.88$ | $31.05 \pm 1.12$|
> > |LotteryFL| $89.15 \pm 0.62$| $66.82 \pm 0.11$ | $28.90 \pm 0.22$|
> >
> > The achieved model densities are 50.6%, 50%, and 49.3% for FMNIST, CIFAR-10, and CIFAR-100, respectively. Since it is well known that filter/neuron-wise pruning is less flexible than unstructured pruning, the results show accuracy loss compared to the original SpaFL. However, we can see that the structured SpaFL still outperforms unstructured baselines at the same model density 50%.

---

> > > ### Comment · Reviewer_Ggae · 2023-11-23
> > > **Thank you for your response**
> > >
> > > Thank you for the detailed response. My questions have been satisfactorily answered and therefore I am increasing my score to 6.

---

> > > > ### Author Response · Authors · 2023-11-23
> > > >
> > > > We appreciate the reviewer for raising the score.

---

### Author Response · Authors · 2023-11-22

We appreciate all the reviewers for their constructive feedback and meaningful questions, which helped us improve our manuscript.
Based on the comments from the reviewers, we revised our manuscript (the updated text is colored in blue), including:

**1. Revised Convergence Analysis:**  We updated our Theorem 1 to explicitly show the impact of data heterogeneity on our convergence as suggested by Reviewer Kuri. We also provided a Corollary in Appendix A.2.1 to mitigate the issue of unbounded loss functions in our convergence analysis as suggested by Reviewer R775.

**2. Ablation Studies:** We added Appendix A.7 and A.8 to show the impact of sparsity regularizer and threshold-based update, respectively, suggested by Reviewer Ggae. We also provided the compatibility of SpaFL with unavailable clients in Appendix A.9. Finally, we showed the performance of SpaFL with ResNet-18 in Appendix A.10

**3. Structured Sparsity:** We added Appendix A.6 to show that SpaFL can be readily extended to perform structured pruning as suggested by Reviewer Ggae.

We also added one more baseline and updated the explanation of Section 3.2.3 and 3.2.4 to enhance the clarity.

---

### Meta-Review · Area_Chair_cgk8 · 2023-12-23

**Metareview:**

(a) The authors propose a new formulation for what they claim is personalized federated learning of sparse models. They also proposed an algorithm to solve it, give a theoretical analysis and conduct experiments, where the claim is that their method outperforms certain baselines.

(b) I believe the formulation is fundamentally flawed and hence I do not think this paper can be described as having any particular strengths.

(c) The key weakness is that their formulation (1) is fundamentally flawed. Before explaining why this is the case, let me fix and simplify certain issues with (1). First, the text should say that $D_k$ is the number of data samples stored on client $k$, and the third line should list $(x_{k,i}, y_{k,i})$ instead of $(x_i, y_i)$, since the data differs across the clients. Second, the notation involving function $F_k$ is unnecessarily complicated. Indeed, it's enough to write

$$F_k(w \circ p_k) = \frac{1}{D_k}\sum_{i=1}^{D_k} {\cal L}(w \circ p_k; (x_{k,i}, y_{k,i}))$$

instead of $F_k(w \circ p_k; p_k)$.

Moreover, it's better to define
$w_k(p_k) := \arg \min_w F_k(w \circ p_k)$ and then replace $w_k^*$ with $w_k(p_k)$. This is because $w_k^*$ clearly depends on $k$ and $p_k$. With these fixes, formulation (1) becomes

$$\min_{p_1, \dots, p_N \in B} \sum_{k=1}^N F_k(w_k(p_k) \circ p_k),$$

where $B$ is the two-element set composed of $0$ and $1$. Notice that this problem is fully separable into $N$ independent problems:

$$ \min_{p_k \in B} F_k(w_k(p_k) \circ p_k), \qquad \text{for} \qquad k=1,2,\dots,N.$$

Moreover, it is easy to observe that these problems are equivalent to

$$\min_{v_k \in R^d} F_k(v_k),  \qquad \text{for} \qquad k=1,2,\dots,N.$$

Indeed, one can simply solve for optimal $v_k$, find its sparsity pattern $p_k(v_k) := (p_k^1, \dots, p_k^d)$ (that is: $p_k^i = 0 $ if $v_k^i=0$ and $p_k^i = 1 $ if $v_k^i \neq 0$), and then the solution to

$$ \min_{p_k \in B} F_k(w_k(p_k) \circ p_k), \qquad \text{for} \qquad k=1,2,\dots,N.$$

is simply $p_k = p_k(v_k)$ and $w_k(p_k) = v_k$. So, problem (1) can be solved with zero (!!!) communication complexity by simply asking each client to minimize their own loss. The rest of the paper is therefore irrelevant, and the theoretical results and numerical experiments vacuous. It may be the case that the idea has some merit, but not before a major revision addressing this fatal issue.

I am surprised that none of the reviewers noticed this. The reviewers noticed some other more minor issues though (some of which were addressed by the authors and some of which were not).

In summary, it is impossible to allow a paper with this kind of a fatal flaw to be accepted.

**Justification For Why Not Higher Score:**

Fatal flaw.

**Justification For Why Not Lower Score:**

N/A

---

### Decision · Program_Chairs · 2024-01-16

Reject